# MergePRAG: Orthogonal Merging of Passage-experts for Multi-hop Parametric RAG

**Xuebing Liu[1], Shanbao Qiao[1], Roseline Nyange[1],**
**Dongwook Min[2], Hyun Kim[3], Seung-Hoon Na[2]** *

[1]Department of Computer Science and Artificial Intelligence, Jeonbuk National University
[2]Graduate School of Artificial Intelligence, Ulsan National Institute of Science and Technology
[3]Electronics and Telecommunications Research Institute
{liuxuebing,joe,roselinenyange}@jbnu.ac.kr
mindw96@unist.ac.kr, h.kim@etri.re.kr, nash@unist.ac.kr

## Abstract

Large language models (LLMs) can be enhanced with external knowledge through two dominant approaches: (1) **retrieval-augmented generation (RAG)**, which supplements LLMs with in-context retrieved passages, and (2) **parametric knowledge adaptation (PKA)**, which directly updates model parameters with new domain knowledge. Recently, parametric RAG (PRAG) has emerged as a promising framework, extending RAG by translating retrieved passages into parameter updates, thereby mitigating inefficiency and noise sensitivity inherent to RAG. However, existing PRAG methods remain limited to single-pass retrieval, falling short of the **multi-hop RAG** setting that requires iterative retrieval and reasoning. We propose **MergePRAG** (*Orthogonal Merging of Passage-experts for Multi-hop PRAG*), a novel framework that sequentially integrates retrieved passages into LLM parameters through a continual merging mechanism, which is advanced by two key proposals: (1) **orthogonal merging** using the Gram–Schmidt process to minimize conflicts between "passage experts", and (2) **critical-layer parameterization** to efficiently encode in-context passages. Experiments on multi-hop open-domain QA and reasoning-aware knowledge editing show that MergePRAG consistently outperforms both standard and state-of-the-art RAGs as well as existing parametric adaptation methods, achieving superior effectiveness and efficiency. All datasets and code will be released at https://github.com/Liu-Xuebing/MhQA_hypernetwork.

## 1 Introduction

Large language models (LLMs)(Dubey et al., 2024; Mesnard et al., 2024; Team, 2024; DeepSeek-AI, 2024) have achieved strong performance on a wide range of knowledge-intensive tasks, driven by billions of parameters and large-scale pretraining corpora. However, their parametric knowledge remains static, making them ill-suited for evolving world knowledge or emerging domains. Retrieval-augmented generation (RAG) has become a popular remedy, injecting retrieved passages into the input context at inference time. While effective, RAG faces challenging issues: (1) *knowledge conflict* between parametric and retrieved information (Xie et al., 2023; Kortukov et al., 2024; Zhang et al., 2025; Bi et al., 2025), (2) inference inefficiency from processing long retrieval-heavy contexts (Leng et al., 2024; Jin et al., 2024; Chen et al., 2025), and (3) noise sensitivity, where irrelevant or erroneous passages degrade performance (Cuconasu et al., 2024; Wu et al., 2024; Fang et al., 2024).

Alternatively, **Parametric RAG (PRAG)**, along with its dynamic variant (Su et al., 2025; Tan et al., 2025a), has recently emerged as a promising direction. PRAG translates retrieved passages into LoRA parameter updates via a "hypernetwork", enabling LLMs to *internalize* external knowledge

---

*Corresponding author

beyond mere in-context conditioning.[1] Notably, PRAG has been shown to consistently outperform standard RAG, both when applied independently and when combined with retrieval-based methods.

Despite its promise, **PRAG has thus far been investigated only in simplified RAG settings**, typically limited to a single retrieval step rather than the more challenging multi-hop RAG scenario (Yu et al., 2024b; Li et al., 2025b). In multi-hop RAG, a complex query is decomposed into subquestions, each requiring iterative retrieval and sub-answer generation, such that retrieved passages are incrementally provided during the question answering (QA) process. A central research question, therefore, is how to effectively extend PRAG to multi-hop settings—where the internalization of retrieved passages must continuously progress across hops—without necessitating the retraining or rebuilding of a hypernetwork originally designed for single-hop RAG. This extension of PRAG to multi-hop RAG represents an important milestone, as it provides a natural bridge toward recent *reasoning-enhanced* RAG frameworks (e.g., IRCoT, Self-RAG, DeepRAG, and RAG-R1 (Trivedi et al., 2023; Asai et al., 2024; Guan et al., 2025; Tan et al., 2025b)).

We propose **MergePRAG** (Orthogonal Merging Passage-experts for Multi-hop PRAG), a generalized framework that scales PRAG to multi-hop RAG. At each stage, retrieved passages are translated into expert parameters by a hypernetwork and merged with the previously accumulated experts through a continual merging mechanism, thus enabling effective accumulation of knowledge across iterative retrievals (Figure 1). For effective continual merging, we propose two advances: (1) **orthogonal merging** using the Gram–Schmidt process to minimize conflicts between newly introduced and existing experts by eliminating redundant information and promoting complementary, non-overlapping representations, and (2) a **critical-layer parameterization** module that updates only the preselected *critical layer* to efficiently encode in-context passages, while keeping the remaining layers frozen to reduce computation and stabilize reasoning. Together, these techniques allow MergePRAG to reuse a single passage-level hypernetwork across hops, avoiding the redesign or retraining of additional hypernetworks to support multi-hop RAG.

Our contributions are threefold: (1) We introduce MergePRAG, the first generalized PRAG framework for multi-hop RAG. (2) We propose a continual merging mechanism that sequentially integrates retrieved passages into LLM parameters, enabled by two advances: orthogonal merging and critical-layer parameterization. (3) We conduct extensive experiments across multiple LLM backbones and benchmark datasets, showing that MergePRAG consistently outperforms existing RAG and PRAG baselines in both effectiveness and efficiency.

## 2 RELATED WORKS

### 2.1 PARAMETRIC KNOWLEDGE ENHANCEMENT

Parametric knowledge enhancement methods aim to increase the knowledge capacity of language models by adjusting their parameters to better encode new information. The most direct approach is *full fine-tuning*, but this quickly becomes impractical as model sizes grow. To address scalability, *parameter-efficient fine-tuning (PEFT)* techniques, such as LoRA (Hu et al., 2021) and its variants (Valipour et al., 2022; Yu et al., 2024a; Liu et al., 2024), update only a small set of low-rank matrices, achieving performance comparable to full fine-tuning at a fraction of the cost. PEFT methods can be attached or removed with minimal disruption, enabling efficient adaptation across domains and tasks. Moreover, they reduce training-time memory and compute requirements.

With the rise of *model editing*, more targeted approaches have been developed that directly locate and modify knowledge representations within the model. Methods such as ROME (Meng et al., 2022a), MEMIT (Tan et al., 2023), and PMET (Li et al., 2024) update critical layers to encode new facts, while MEND (Mitchell et al., 2021) and MALMEN (Tan et al., 2023) employ hypernetworks to inject knowledge into specific layers, effectively fusing edits with existing parameters. To mitigate catastrophic forgetting and preserve general-purpose capabilities, approaches like T-Patcher (Huang et al., 2023) and MEMoE (Wang & Li, 2024) introduce external memory modules that store edits separately from the core model. These designs are particularly valuable in dynamic settings where

---

[1] In this paper, we use PRAG as a broad term encompassing the original PRAG (Su et al., 2025) and its variants, including DyPRAG (Tan et al., 2025a).

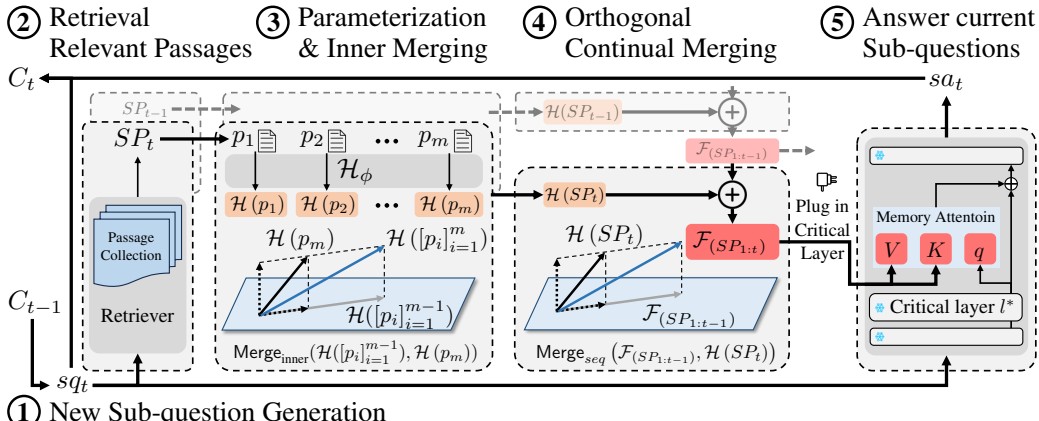

Figure 1: Overview of MergePRAG for multi-hop QA. A complex query is decomposed into sub-questions, and retrieved passages are sequentially incorporated through parameterization and continual merging. At each timestep $t$: (1) a sub-question $sq_t$ is generated from the reasoning chain $C_{t-1}$ (Eq. 1, Section 3.1); (2) the retriever returns top-ranked passages $SP_t \subseteq \mathcal{R}$; (3) given $SP_t = [p_i]_{i=1}^m$, each passage is parameterized by the hypernetwork to produce $\{\mathcal{H}_\phi(p_i)\}_{i=1}^m$, which are combined into $\mathcal{H}_\phi(SP_t)$ via the inner-merging mechanism (Eq. 6, Section 3.2); (4) orthogonal continual merging updates the accumulated parameters $\mathcal{F}(SP_{1:t-1})$ with $\mathcal{H}_\phi(SP_t)$ to obtain $\mathcal{F}(SP_{1:t})$ (Eq. 11, Section 3.2.2); and (5) the merged expert $\mathcal{F}(SP_{1:t})$ is injected into the base LLM $\mathcal{M}_{\theta_0}$ at the critical layer $l^*$ to generate the sub-answer (Eqs. 4–5). This process repeats until no further sub-questions are produced, after which the final answer is generated.

facts evolve over time and updates must be applied repeatedly. They also support more controllable interventions by localizing changes to a small subset of parameters or auxiliary components.

Overall, parametric enhancement methods differ in *where* and *how* they modify parameters—ranging from full updates to low-rank adapters, targeted edits, or external memory—yet they share the goal of augmenting LLMs with new knowledge while retaining general abilities.

## 2.2 RETRIEVAL AUGMENTED GENERATION

Early RAG methods (Lewis et al., 2020; Guu et al., 2020; Izacard & Grave, 2021; Borgeaud et al., 2022) train language models jointly with top-retrieved documents, enabling the model to incorporate external knowledge sources when generating answers. To further improve performance, subsequent approaches expanded the knowledge sources, incorporated query rewriting, or jointly trained the retriever and the generator to achieve tighter integration. To mitigate the computational overhead of fully parameterized RAG training, methods such as PRAG (Su et al., 2025) and DyRAG (Tan et al., 2025a) have been proposed, which enhance the model's internal knowledge by learning mappings from retrieved documents to model parameters.

Recent advances increasingly emphasize the importance of *reasoning* over retrieved facts. For instance, FLARE (Jiang et al., 2023), MeLLo (Zhong et al., 2023), IRCoT (Trivedi et al., 2023) and SELF-REASONING (Xia et al., 2025) employ iterative cycles of reasoning, retrieval, and error correction to refine responses. DeepRAG (Guan et al., 2025) formulates reasoning as a Markov Decision Process (MDP) to enable adaptive retrieval, while R3-RAG (Li et al., 2025b) leverages large models to construct trajectories and applies reinforcement learning to teach LLMs stepwise reasoning and retrieval strategies. Collectively, these works highlight the effectiveness of constructing chain-of-thought reasoning processes for complex tasks.

Building on these insights, we present MergePRAG, which extends PRAG to the multi-hop RAG setting and serves as a critical stepping stone toward reasoning-enhanced RAG systems. In contrast to prior PRAG methods (Su et al., 2025) that rely on simple arithmetic merging, MergePRAG introduces a merging module with orthogonal merging, enabling more effective integration of passage experts across hops.

## 3 METHODOLOGY

In this section, we present MergePRAG, illustrated in Figure 1. We first provide a brief background on multi-hop RAG, and then describe MergePRAG and its two main components: orthogonal merging with the Gram–Schmidt process and critical-layer parameterization.

We define two language models and a retrieval module. $\mathcal{M}_{\theta_0}$ denotes a *general-purpose LLM* for sub-answer generation, also referred to as the *base LM*, $\mathcal{M}_{sq}$ a *sub-question generator*, based a smaller LLM, and $\mathcal{R}$ the *retriever*, which returns a set of top-ranked passages for each query $q$, denoted as $\mathcal{R}(q)$.

### 3.1 MULTI-HOP RAG

Let $q$ be the original complex query. In the multi-hop RAG setting, each step involves sub-question generation, retrieval, and response generation. At step $t$, given $C_{t-1}$, the accumulated context so far, the next sub-question $sq_t$ and its sub-answer $sa_t$ are obtained as

$$sq_t = \mathcal{M}_{sq}(C_{t-1}), sa_t = \mathcal{M}_{\theta_0}(sq_t, SP_t), \quad SP_t \subseteq \mathcal{R}(sq_t), \tag{1}$$

where $SP_t$ denotes the retrieved passages at step $t$. Task-specific instruction prompts for $\mathcal{M}_{sq}$ and $\mathcal{M}_{\theta_0}$ are described in Appendix H.

The newly obtained tuple $(sq_t, sa_t)$ is appended to the context: $C_t = [C_{t-1}, sq_t, sa_t]$. The final answer to the original query $q$ is then generated as $a = \mathcal{M}(C_{T-1}, q)$, where $sq_T = \mathcal{M}_{sq}(C_{T-1}) = \langle \text{EOS} \rangle$.

In the single-hop setting, RAG produces the answer in one step: $a = \mathcal{M}_{\theta_0}(q, SP_1), \quad SP_1 \subseteq \mathcal{R}(q)$, and the process terminates immediately.

### 3.2 MERGEPRAG

To present MergePRAG, we first review PRAG in the single-hop RAG setting.

**PRAG.** As in DyPRAG (Tan et al., 2025a), PRAG employs a hypernetwork-based passage parameterization module. Let $\mathcal{H}_\phi$ denote the hypernetwork, which maps a retrieved passage $p$ to a set of passage-specific LoRA parameters:$\theta_p = \mathcal{H}_\phi(p)$. The hypernetwork is trained to efficiently translate an in-context passage into its corresponding parameters.

Let $\mathcal{M}_{\theta_0}$ denote the base LLM with parameters $\theta_0$. PRAG augments the model by injecting *passage-specific parameters* $\mathcal{H}_\phi(p)$ generated from the hypernetwork $\mathcal{H}_\phi$, such that for a passage $p$,

$$\theta' = \theta_0 \oplus \mathcal{H}_\phi(p),$$

where $\oplus$ denotes the *parameter-injection* operation. Unlike RAG, which conditions on the passage $p$ explicitly in the input prompt, PRAG generates the answer under the passage-injected model as

$$a = \mathcal{M}_{\theta_0 \oplus \mathcal{H}_\phi(p)}(q). \tag{2}$$

**MergePRAG.** MergePRAG extends PRAG to the multi-hop RAG setting, where passages arrive sequentially through iterative retrieval. By timestep $t$, the accumulated passages are $SP_{1:t} = [SP_1, \ldots, SP_t]$. To inject all context passages into the LLM parameters, let $\mathcal{F}$ denote a mapping from the sequence $SP_{1:t}$ to the parameter space. Instead of directly "training" $\mathcal{F}$ over datasets with varying numbers of passages $t$, **MergePRAG introduces a continual merging mechanism that induces $\mathcal{F}$ by reusing the *passage-level hypernetwork* $\mathcal{H}_\phi$**, which maps a single passage to its parameter representation.

**Sequence-merging.** The sequence merging, denoted as $\text{Merge}_{seq}$, is a recursive operation that combines the previously accumulated parameters $\mathcal{F}_{(SP_{1:t-1})}$ with the new passage-specific parameters $\mathcal{H}_\phi(SP_t)$:

$$\mathcal{F}_{(SP_{1:t})} = \text{Merge}_{seq}\big(\mathcal{F}_{(SP_{1:t-1})}, \mathcal{H}_\phi(SP_t)\big). \tag{3}$$

Using the "merged" parameter representation, MergePRAG generates a candidate answer at timestep $t$ without relying on in-context passages:

$$sa_t = \mathcal{M}_{\theta_0 \oplus \mathcal{F}_{(SP_{1:t})}}\big(sq_t\big), \tag{4}$$

At the final timestep $T$, MergePRAG generates the final answer as $a = \mathcal{M}_{\theta_0 \oplus \mathcal{F}_{(SP_{1:T})}}(q)$.

**MergePRAG+.** Similar to PRAG-Combine (Su et al., 2025), **MergePRAG+** integrates RAG and PRAG in a complementary manner, yielding:

$$
\begin{aligned}
sa_t &= \mathcal{M}_{\theta_0 \oplus \mathcal{F}_{(SP_{1:t})}}(SP_t, sq_t), \quad t < T, \\
a &= \mathcal{M}_{\theta_0 \oplus \mathcal{F}_{(SP_{1:T})}}(C_T, q), \quad t = T.
\end{aligned} \tag{5}
$$

**Inner-merging.** We introduce an *inner-merging* mechanism to induce $\mathcal{H}(SP)$ from individual passage parameters, for $|SP| > 1$. Formally, given a list of passages $SP = [p_1, \ldots, p_m]$, $\mathcal{H}(SP)$ is obtained by applying an inner merging operation $\text{Merge}_{\text{inner}}$:

$$
\begin{aligned}
\mathcal{H}([p_i]_{i=1}^m) &= \text{Merge}_{\text{inner}}(\mathcal{H}_\phi(p_1), \ldots, \mathcal{H}_\phi(p_m)) \\
&= \text{Merge}_{\text{inner}}\left(\mathcal{H}([p_i]_{i=1}^{m-1}), \mathcal{H}(p_m)\right)
\end{aligned} \tag{6}
$$

### 3.2.1 HYPERNETWORK-BASED KEY–VALUE MEMORY PARAMETERIZATION FOR $\mathcal{H}_\phi$.

For passage parameterization, MergePRAG adopts a key–value memory parameterization scheme, where the hypernetwork generates $k$ key and value vectors for each passage, which serve as a "compressed" *passage-specific memory*. The passage-specific memory is inserted into the feed-forward network (FFN) at the critical layer $l^*$ via an additional attention mechanism, referred to as the *memory attention* mechanism.

Formally, the hypernetwork $\mathcal{H}_\phi(p)$ first produces the passage-specific memory for passage $p$ as:

$$
\mathcal{H}_\phi(p) = \{\mathbf{K}_p, \mathbf{V}_p\}, \tag{7}
$$

where $\mathbf{K}_p, \mathbf{V}_p \in \mathbb{R}^{k \times d_{\text{out}}}$ are the key and value matrices, respectively.

Suppose that the original FFN module at layer $l^*$ is denoted as a function $\text{MLP}_{\theta_0} : \mathbb{R}^{d_{\text{in}}} \to \mathbb{R}^{d_{\text{out}}}$ parameterized by $\theta_0$. The passage-specific FFN expert $E_{\mathcal{H}_\phi(p)}$ is then obtained for an input $\mathbf{x} \in \mathbb{R}^{d_{\text{in}}}$ using a memory attention mechanism, i.e., standard attention applied to the passage-specific memory $(\mathbf{K}_p, \mathbf{V}_p)$ with the base FFN output $\text{MLP}_{\theta_0}(\mathbf{x})$ used as the query. Formally,

$$
\begin{aligned}
E_{\mathcal{H}_\phi(p)}(\mathbf{x}) &= \text{Attention}(\text{MLP}_{\theta_0}(\mathbf{x}), \, \mathbf{K}_p, \, \mathbf{V}_p), \\
\text{Attention}(\mathbf{q}, \mathbf{K}_p, \mathbf{V}_p) &= \text{softmax}\left(\frac{\mathbf{q}\mathbf{K}_p^\top}{\sqrt{d_{\text{out}}}}\right)\mathbf{V}_p,
\end{aligned} \tag{8}
$$

The passage-specific FFN expert is injected into the original FFN layer at $l^*$, yielding:

$$
\text{MLP}_{\theta_0 \oplus \mathcal{H}_\phi(p)}(\mathbf{x}) = \text{MLP}_{\theta_0}(\mathbf{x}) + E_{\mathcal{H}_\phi(p)}(\mathbf{x}). \tag{9}
$$

### 3.2.2 ORTHOGONAL CONTINUAL MERGING MECHANISM (Merge) FOR $\mathcal{F}$

Once the parameterization module $\mathcal{H}_\phi(SP_i)$ produces passage vectors $(\mathbf{K}_p, \mathbf{V}_p)$ as in Eq. 7, the continual merging mechanism operates on each parameter independently. To form a merged expert without overwriting previously acquired knowledge, we propose an *orthogonal merging* method based on Gram–Schmidt projection, inspired by recent studies (Xu et al., 2025). Formally, let $\{\mathbf{W}_i\}_{i=1}^t$ denote the set of key or value memory matrices (i.e., $\mathbf{K}_p$ or $\mathbf{V}_p$) obtained from $\{\mathcal{H}_\phi(SP_i)\}_{i=1}^t$, where $\mathbf{W}_i \in \mathbb{R}^{k \times d_{out}}$.

Let $\mathbf{W}_{\mathcal{F}}^{t-1}$ be the merged parameter obtained from $\{\mathbf{W}_i\}_{i=1}^{t-1}$ up to step $t-1$. The Gram–Schmidt orthogonalization procedure first computes the projection matrix onto the subspace spanned by $\mathbf{W}_{\mathcal{F}}^{t-1}$:

$$
\mathbf{P}^{t-1} = \mathbf{W}_{\mathcal{F}}^{t-1}\left((\mathbf{W}_{\mathcal{F}}^{t-1})^\top \mathbf{W}_{\mathcal{F}}^{t-1}\right)^{-1}(\mathbf{W}_{\mathcal{F}}^{t-1})^\top. \tag{10}
$$

The new parameter $\mathbf{W}_t$ is then merged by adding only its orthogonal component with respect to the subspace spanned by $\mathbf{W}_{\mathcal{F}}^{t-1}$:

$$
\mathbf{W}_{\mathcal{F}}^t = \mathbf{W}_{\mathcal{F}}^{t-1} + \left(\mathbf{I} - \mathbf{P}^{t-1}\right)\mathbf{W}_t, \tag{11}
$$

where $\mathbf{P}^{t-1}$ is the projection matrix defined in Eq. 10. A detailed discussion of orthogonal merging using the Gram–Schmidt procedure is provided in Appendix B.

### 3.2.3 HYPERNETWORK ARCHITECTURE: SEQUENCE-TO-MEMORY

The hypernetwork is designed to take a token sequence of a passage and produce its key–value memory. Given a passage as an input sequence of tokens, the hypernetwork $\mathcal{H}_\phi$ first computes a passage embedding via *attentive pooling* over the token-level embeddings. The resulting passage embedding is then passed through a two-layer MLP, whose output is transformed by linear projections to generate the passage-specific memory, i.e., the key and value matrices.

Formally, given a passage $p$, we denote its sentence embedding by $\mathsf{Emd}(p)$, obtained as the attentively pooled representation from an auxiliary Transformer encoder (Appendix C). The hypernetwork then transforms $\mathsf{Emd}(p)$ into a latent representation using $\mathsf{MLP}_{\text{hyp}}$, as follows:

$$\mathbf{h}_b = \mathsf{MLP}_{\text{hyp}}(\mathsf{Emd}(p)) = \mathrm{ReLU}(\mathbf{V}' \, \mathrm{LN}\,(\mathrm{ReLU}(\mathbf{W}' \, \mathsf{Emd}(p)))). \tag{12}$$

where LN refers to the layer normalization layer.

Finally, we apply two distinct linear transformations to map the latent representation $\mathbf{h}_b$ into flattened key and value matrices, i.e., the "passage-specific memory" for $p$:

$$\mathbf{K}_p = \mathbf{W}_K \, \mathbf{h}_b + \mathbf{b}_K, \quad \mathbf{V}_p = \mathbf{W}_V \, \mathbf{h}_b + \mathbf{b}_V, \tag{13}$$

where $\mathbf{W}_K, \mathbf{W}_V \in \mathbb{R}^{K \times d \times d_{\text{hid}}}$ are linear projection tensors and $\mathbf{b}_K, \mathbf{b}_V \in \mathbb{R}^{K \times d}$ are bias terms. With a slight abuse of notation, we treat a matrix in $\mathbb{R}^{K \times d \times 1}$ as a matrix in $\mathbb{R}^{K \times d}$ by removing the singleton dimension. More details of the hypernetwork architecture are provided in Appendix C.

### 3.2.4 CRITICAL-LAYER PARAMETERIZATION FOR $\mathcal{H}_\phi$

**The *critical-layer* parameterization applies $\mathcal{H}$ only to a single critical layer $l^*$**, rather than across all layers, motivated by the locate-and-edit methods of (Meng et al., 2022a;b; Li et al., 2024; Fang et al., 2025).

To identify the critical layer $l^*$, we conduct layer-wise scanning experiments on both models across all datasets. For each layer, we measure the change in perplexity after injecting the corresponding passage vectors, thereby evaluating the effectiveness of the layer-specific hypernetwork (see Appendix: A). As shown in Figure 2 and 3, the early-to-middle layers contribute most substantially when used as parameterization modules. Based on this analysis, the insertion positions for the single-layer passage-vector parameterization are summarized in Table 8.

### 3.2.5 TRAINING OBJECTIVE

**Hypernetwork $\mathcal{H}_\phi$.** To train $\mathcal{H}_\phi$,[2] we construct a dataset $\mathcal{D}_\mathcal{H} = \{(q_i, p_i, a_i)\}_{i=1}^N$, where each triple consists of a question $q_i$, its relevant passage $p_i$, and the ground-truth answer $a_i$. The hypernetwork is trained by minimizing the cross-entropy loss:

$$\mathcal{L}_{\text{CE}}(\phi) = - \sum_{(q,p,a) \in \mathcal{D}_\mathcal{H}} \log P_{\mathcal{M}_{\theta_0 \oplus \mathcal{H}_\phi(p)}}(a \mid q), \tag{14}$$

where $P_{\mathcal{M}_{\theta_0 \oplus \mathcal{H}_\phi(p)}}(a \mid q)$ denotes the probability of generating answer $a$ conditioned on question $q$ under the parameters of the passage-injected model $\mathcal{M}_{\theta_0 \oplus \mathcal{H}_\phi(p)}$.

**Subquestion generator $\mathcal{M}_{sq}$.** Following Li et al. (2025b), we adopt a cold-start stage to train the sub-question generator $\mathcal{M}_{sq}$ by constructing a dataset $\mathcal{D}_{sq} = \{(q^{(j)}, y^{(j)})\}_{j=1}^M$, where each target sequence is

$$y^{(j)} = [\, sq_1^{(j)}, sa_1^{(j)}, sq_2^{(j)}, sa_2^{(j)}, \dots, sq_{n_j}^{(j)}, sa_{n_j}^{(j)}, \langle \text{EOS} \rangle \,].$$

The autoregressive objective on $\mathcal{D}_{sq}$ is used to train $\mathcal{M}_{sq}$, as detailed in Appendix D.

## 4 EXPERIMENTS

### 4.1 EXPERIMENTS SETTING

**Models and Datasets.** We employ LLaMA3.1-8B (Dubey et al., 2024) and Qwen2.5-7B (Team, 2024) as research base models. For the multi-hop question answering task, we follow works (Guan

---

[2]Here, $\mathcal{H}_\phi$ denotes the layer-specific hypernetwork that injects passage knowledge into the FNN at the critical layer $l^*$.

et al., 2025; Li et al., 2025b) and utilize the E5 (Wang et al., 2022) and BM25 (Lù, 2024) retrievers. For the multi-hop editing task, we follow work (Zhong et al., 2023) and adopt the Contriever model (Lei et al., 2023) as the retriever. [3]

We conduct experiments on multi-hop question answering benchmarks, including HotpotQA (Yang et al., 2018), 2WikiMultihopQA (2WikiMhQA) (Ho et al., 2020), and MuSiQue (Trivedi et al., 2022). In addition, we evaluate multi-hop editing on the MQuAKE (Zhong et al., 2023) dataset, with detailed experimental settings and results reported in Appendix E.1.

**Metrics.** We evaluate model performance using Exact Match (EM) and F1 score (F1) (Kwiatkowski et al., 2019). EM measures the strict string-level agreement between predictions and gold answers, while F1 quantifies partial correctness by computing the token-level overlap between predictions and references. For all experiments, we take the model's final response as its predicted answer and compare it against the gold standard.

**Baselines.** We evaluate our approach against a range of baselines: (i) RAG and RAG-CoT, which retrieve relevant documents to answer queries, (ii) iterative retrieval methods such as IRCoT, FLARE and MeLLo, (iii) parameterized RAG methods including PRAG and DyPRAG and (iv) reasoning-enhanced RAG methods including Auto-RAG, Adaptive-RAG, Deep-RAG, R3-RAG, Search-R1 and Search-o1. The detailed descriptions of these baseline methods can be found in the Appendix F.

**Implementation Details.** All experiments were conducted on a workstation with 8 NVIDIA RTX A6000 GPUs. The detailed training settings and inference are provided in Appendix D.

## 4.2 MAIN RESULTS AND ANALYSIS

We evaluated MergePRAG on multi-hop QA datasets using LLaMA3.1-8B and Qwen2.5-7B, with results summarized in Table 1. MergePRAG consistently outperforms state-of-the-art baselines across all three datasets, showing the best performances in most cases, except for the run using LLaMA3.1-8B on MuSiQue. Compared with early passage-injection methods such as PRAG and DyPRAG, MergePRAG+ achieves higher performance, demonstrating that the hypernetwork-based parameterization framework extends effectively to multi-hop QA. Additional gains are obtained when combined with explicit in-context passages, without sacrificing generalization. The results further indicate that increasing the number of retrieved passages $|SP|$ with $\text{Merge}_{\text{inner}}$ provides additional improvements over using a single passage ($|SP| = 1$).

To examine the effect of hypernetwork-based parameterization, we include an additional baseline, **MultihopRAG** (Section 3.1), which directly uses the original LLM $\theta_0$ without hypernetwork-based parameterization or injection (Algorithm 2). Comparisons with MultihopRAG show that hypernetwork-based passage knowledge injection contributes substantially to performance gains.

## 4.3 ABLATION STUDY

We conducted a series of ablation studies to examine the effectiveness of the proposed framework and to identify the contribution of its key components. In addition, we performed efficiency analysis experiments to evaluate the computational performance of our approach; the detailed results are presented in Appendix E.2.

### 4.3.1 MERGEPRAG+ VS. MULTIHOPRAG W/ FINETUNING

To compare standard *fine-tuning* with the proposed hypernetwork-based parameterization in MergePRAG, we apply fine-tuning to **MultihopRAG**, directly adjusting $\theta_0$ on the same training data used in our framework. We consider two settings: (1) *fine-tuning without passages*, i.e., $[sq \rightarrow sa]$, where the model is trained to predict $sa$ from $sq$ alone; and (2) *fine-tuning with passages*, i.e., $[(P_{\text{gold}}, sq) \rightarrow sa]$, where the model is trained to predict $sa$ given $sq$ and the gold passages, resembling the standard RAG training paradigm.

---

[3]We follow these works for a fair comparison. Pre-trained models can be obtained from Hugging Face.
LLaMA-3.1-8B: `https://huggingface.co/meta-llama/Llama-3.1-8B`
Qwen2.5-7B: `https://huggingface.co/Qwen/Qwen2.5-7B`
E5: `https://huggingface.co/intfloat/e5-base-v2`
Contriever: `https://huggingface.co/facebook/contriever-msmarco`

| Model | Retriever | Method | HotpotQA | | 2WikiMhQA | | MuSiQue | |
|---|---|---|---|---|---|---|---|---|
| | | | EM | F1 | EM | F1 | EM | F1 |
| LLaMA3.1-8B | E5 | RAG$_{|SP|=1}$ | 21.60 | 36.67 | 4.90 | 17.36 | 2.00 | 11.49 |
| | E5 | RAG$_{|SP|=4}$ | 27.80 | 40.51 | 5.10 | 15.80 | 2.70 | 11.27 |
| | E5 | RAG-CoT$_{|SP|=1}$ | 37.60 | 45.15 | 30.90 | 35.00 | 5.60 | 13.38 |
| | E5 | RAG-CoT$_{|SP|=4}$ | 43.70 | 50.41 | 36.20 | 40.00 | 5.90 | 12.49 |
| | E5 | IRCoT[†] | 39.30 | 46.00 | 35.10 | 37.50 | 12.00 | 13.60 |
| | E5 | FLARE[†] | 17.80 | 20.90 | 10.90 | 11.40 | 2.30 | 2.80 |
| | E5 | R3-RAG[†] | 45.60 | 58.80 | 52.90 | 60.90 | **21.20** | **32.80** |
| | BM25 | R3-RAG[†] | 44.40 | 57.60 | 50.60 | 58.60 | 17.20 | 27.70 |
| | BM25 | Search-o1[†] | 14.80 | 24.08 | 22.20 | 27.10 | 5.40 | 11.98 |
| | BM25 | Auto-RAG[†] | 25.80 | 36.09 | 23.00 | 30.09 | - | - |
| | BM25 | DeepRAG[†] | 40.70 | 51.54 | 48.10 | 53.25 | - | - |
| | BM25 | PRAG[†] | - | 44.84 | - | 40.55 | - | - |
| | BM25 | DyPRAG[†] | - | 38.35 | - | 50.24 | - | - |
| | E5 | **MergePRAG +**$_{|SP|=1}$ | 48.80 | 55.53 | 66.30 | 71.05 | 14.40 | 25.04 |
| | E5 | **MergePRAG +**$_{|SP|=4}$ | **52.40** | **60.67** | **73.20** | **79.34** | 16.70 | 27.69 |
| | BM25 | **MergePRAG +**$_{|SP|=1}$ | 46.80 | 53.40 | 61.10 | 67.31 | 17.80 | 29.39 |
| | BM25 | **MergePRAG +**$_{|SP|=4}$ | 52.40 | 60.58 | 70.20 | 76.65 | 20.30 | 31.20 |
| Qwen2.5-7B | E5 | RAG$_{|SP|=1}$ | 36.60 | 43.37 | 34.90 | 37.36 | 3.20 | 8.71 |
| | E5 | RAG$_{|SP|=4}$ | 45.30 | 52.08 | 42.00 | 44.49 | 5.80 | 12.73 |
| | E5 | RAG-CoT$_{|SP|=1}$ | 30.20 | 36.20 | 19.10 | 23.05 | 4.30 | 8.30 |
| | E5 | RAG-CoT$_{|SP|=4}$ | 44.60 | 51.28 | 35.40 | 37.79 | 5.20 | 9.55 |
| | E5 | IRCoT[†] | 35.70 | 41.10 | 31.10 | 33.50 | 9.40 | 11.20 |
| | E5 | FLARE[†] | 23.40 | 32.06 | 21.80 | 26.51 | 3.60 | 4.80 |
| | E5 | R3-RAG[†] | 46.40 | **59.70** | 54.20 | 62.70 | **21.40** | **34.00** |
| | BM25 | R3-RAG[†] | 44.90 | 58.20 | 52.80 | 61.10 | 17.60 | 30.00 |
| | BM25 | Search-o1[†] | 11.60 | 16.95 | 22.00 | 25.02 | 2.10 | 7.48 |
| | BM25 | DeepRAG[†] | 32.10 | 41.14 | 40.40 | 44.87 | - | - |
| | E5 | **MergePRAG +**$_{|SP|=1}$ | 43.40 | 50.64 | 65.80 | 69.72 | 9.70 | 19.61 |
| | E5 | **MergePRAG +**$_{|SP|=4}$ | 50.80 | 58.37 | **77.40** | **81.49** | 12.30 | 21.57 |
| | BM25 | **MergePRAG +**$_{|SP|=1}$ | 42.00 | 49.09 | 59.70 | 63.05 | 13.00 | 23.35 |
| | BM25 | **MergePRAG +**$_{|SP|=4}$ | **51.40** | 59.33 | 71.80 | 76.06 | 16.70 | 27.33 |

Table 1: Overall results on three multi-hop QA tasks. Bold numbers indicate the best performance. [†] denotes results reported from the original papers or R3-RAG paper. PRAG and DyPRAG results correspond to the combined setting with in-context passages (i.e., PRAG-Combine and DyPRAG-Combine). In MergePRAG runs, $|SP|$ refers to the number of retrieved passages per hop. MergePRAG applies orthogonal continual merging (Section 3.2.2) for both inner-merging and sequence-merging, i.e., Merge$_{inner}$ and Merge$_{seq}$. Additional results obtained using alternative models and methods are provided in Table 12.

| Traing type | HotpotQA | | 2WikiMhQA | |
|---|---|---|---|---|
| | EM | F1 | EM | F1 |
| MultihopRAG$_{|SP|=1}$ (w/o finetuning) | 37.80 | 47.56 | 23.30 | 35.59 |
| MultihopRAG$_{|SP|=1}$ (finetuning: $[sq \rightarrow sa]$) | 43.70 | 50.15 | 58.10 | 62.57 |
| MultihopRAG$_{|SP|=1}$ (finetuning: $[(P_{gold}, sq) \rightarrow sa]$) | 40.10 | 46.79 | 60.30 | 62.04 |
| MergePRAG +$_{|SP|=1}$ | 47.40 | 55.29 | 65.60 | 70.54 |

Table 2: Comparison of MergePRAG+ and MultihopRAG with fine-tuning (without hypernetwork) under LLaMA3.1-8B.

Under the LLaMA3.1-8B model with $|SP_i| = 1$, Table 2 compares these MultihopRAG variants with MergePRAG. Interestingly, naive *fine-tuning with passages* ($[(P_{gold}, sq) \rightarrow sa]$) performs even worse than *fine-tuning without passages* ($[sq \rightarrow sa]$). These results are consistent with prior findings (Yang et al., 2024; Lampinen et al., 2025), which show that directly fine-tuning LLMs on domain-adaptive data may degrade their generalization ability.

| Inference type | *HotpotQA* | | 2WikiMhQA | |
|---|---|---|---|---|
| | EM | F1 | EM | F1 |
| $RAG_{|SP|=1}$ | 21.60 | 36.67 | 4.90 | 17.36 |
| $MergePRAG_{|SP|=0}$ | 28.40 | 35.52 | 45.60 | 50.06 |
| $MergePRAG +_{|SP|=1}$ | 48.80 | 55.53 | 66.30 | 71.05 |

Table 3: Comparison of MergePRAG and MergePRAG+ under LLaMA3.1-8B. $|SP| = 0$ denotes MergePRAG, which does not use in-context passages as prompts during inference.

### 4.3.2 MERGEPRAG VS. MERGEPRAG+

Table 3 compares MergePRAG with MergePRAG+. MergePRAG+ exhibits strong generalization and is not negatively affected even when in-context passages are provided. In contrast, applying fine-tuning methods to MultihopRAG leads to performance degradation, implying that direct fine-tuning is unstable for preserving generalization (Section 4.3.1). Overall, these results highlight that MergePRAG preserves the model's ability to perform RAG while benefiting from parameterized knowledge injection, compared with standard fine-tuning methods.

### 4.3.3 EFFECT OF THE MERGING METHODS

To evaluate the effectiveness of the proposed orthogonal merging method in Section 3.2.2, we conduct ablation experiments on HotpotQA using the LLaMA3.1–8B model. Table 5 reports the results of different merging methods for sequence-level merging under the setting $|SP| = 1$, where each sub-question $sq$ retrieves only a single passage. Details of the merging methods are provided in Appendix G.

The results show that the proposed orthogonal merging achieves the best performance, improving by 2.4% over TIES-merging, while arithmetic mean merging also performs comparably. Furthermore, Table 4 presents comparisons using different merging methods for both inner merging $Merge_{inner}$ and inter-merging $Merge_{seq}$ across varying values of $|SP|$. Although arithmetic merging is competitive in most settings, orthogonal merging consistently achieves the best results, often showing an improvement of approximately 1% EM over arithmetic merging. We expect that orthogonal merging will exhibit greater robustness in scenarios with more severe knowledge conflict.

### 4.3.4 EFFECT OF THE NUMBER OF PASSAGES PER RETRIEVAL ($|SP| > 1$)

Table 6 reports the results of MergePRAG+ under different numbers of retrieved passages $|SP|$. As $|SP|$ increases, MergePRAG+ consistently improves performance without degradation, even when longer in-context passages are provided.

### 4.3.5 EFFECT OF THE NUMBER OF KEY–VALUE VECTORS

To examine the impact of the number of key–value vectors used for passage-knowledge parameterization, we conduct an ablation study on HotpotQA and 2WikiMhQA using LLaMA3.1-8B. For each dataset, we train models with different values of $k$ (i.e., $num_{kv}$) under two retrieval settings: $|SP| = 1$ and $|SP| = 4$. The results are summarized in Table 7.

Overall, increasing the number of KV vectors ($k$) consistently improves performance across both datasets and retrieval settings, as a larger $k$ expands memory capacity and preserves more passage-

| $Merge_{inner}$ | $Merge_{seq}$ | $|SP| = 2$ | | $|SP| = 4$ | | $|SP| = 6$ | | $|SP| = 8$ | | $|SP| = 10$ | | $|SP| = 12$ | |
|---|---|---|---|---|---|---|---|---|---|---|---|---|---|
| | | EM | F1 | EM | F1 | EM | F1 | EM | F1 | EM | F1 | EM | F1 |
| ● | ● | 50.20 | 58.07 | 51.40 | 60.35 | 51.40 | 59.82 | 52.00 | 60.86 | 55.00 | 62.84 | 54.40 | 62.76 |
| ● | ■ | 50.60 | 58.06 | 52.00 | 60.64 | 52.00 | 60.21 | 53.00 | 61.32 | 55.40 | 63.40 | 54.60 | 62.64 |
| ■ | ● | 50.40 | 58.14 | 51.60 | 60.13 | 51.40 | 59.63 | 52.60 | 61.46 | 54.80 | 62.80 | 54.60 | 62.76 |
| ■ | ■ | **50.80** | **58.23** | **52.40** | **60.67** | **52.40** | **60.67** | **53.40** | **61.77** | **55.60** | **63.45** | **55.00** | **62.93** |

Table 4: Performance comparison between different merging methods for $Merge_{inner}$ and $Merge_{seq}$ varying $|SP|$: ●: Arithmetic mean merging, ■: Gram–Schmidt orthogonalization merging.

| MergePRAG+ / *HotpotQA* | | |
|---|---|---|
| Merge$_{seq}$ | EM | F1 |
| ▲ | 36.20 | 43.47 |
| ● | 48.20 | 55.04 |
| ♦ | 46.40 | 54.07 |
| ▼ | 48.20 | 54.95 |
| ■ | 48.80 | 55.53 |

Table 5: Performance comparison between different merging methods for Merge$_{seq}$ under the setting of $|SP| = 1$: ▲: Additive merging, ●: Arithmetic mean merging, ♦: TIES merging, ▼: Concat merging, ■: Gram–Schmidt orthogonalization merging.

| | *HotpotQA* | | *2WikiMhQA* | |
|---|---|---|---|---|
| #$|SP|$ | EM | F1 | EM | F1 |
| 1 | 48.80 | 55.53 | 66.30 | 71.05 |
| 2 | 50.80 | 58.23 | 71.40 | 76.94 |
| 3 | 52.00 | 59.50 | 73.10 | 79.06 |
| 4 | 52.40 | 60.67 | 73.20 | 79.34 |

Table 6: Performance of MergePRAG+ on HotpotQA and 2WikiMHQA with varying numbers of retrieved passages ($|SP|$) per sub-question. Increasing $|SP|$ provides broader evidence for answering each sub-question, which can improve overall QA accuracy.

| | *HotpotQA* | | | | *2WikiMhQA* | | | |
|---|---|---|---|---|---|---|---|---|
| | $|SP| = 1$ | | $|SP| = 4$ | | $|SP| = 1$ | | $|SP| = 4$ | |
| $k$ (i.e., #$num_{kv}$) | EM | F1 | EM | F1 | EM | F1 | EM | F1 |
| 1 | 45.60 | 52.67 | 49.00 | 58.24 | 62.40 | 67.89 | 69.00 | 75.21 |
| 2 | 45.60 | 52.67 | 51.20 | 59.20 | 63.80 | 69.01 | 69.00 | 76.32 |
| 4 | 45.60 | 52.86 | 50.80 | 58.88 | 64.00 | 69.37 | 71.20 | 77.09 |
| 8 | 46.40 | 54.25 | 49.40 | 58.39 | 65.90 | 70.93 | 72.00 | 78.09 |
| 16 | 48.80 | 55.53 | 52.40 | 60.67 | 66.30 | 71.05 | 73.20 | 79.34 |

Table 7: Ablation study on the number of passage vectors $k$ (i.e., $num_{kv}$) for LLaMA3.1-8B.

specific information. By capturing richer passage-level representations and mitigating information loss, larger $k$ boosts both EM and F1.

## 5 CONCLUSION

In this work, we introduced MERGEPRAG, which generalizes the PRAG framework to the multi-hop QA setting—an important milestone toward reasoning-enhanced RAG. We proposed two key technical components: (1) *orthogonal continual merging*, which incrementally updates passage experts with newly retrieved knowledge during multi-hop inference while avoiding interference; and (2) *critical-layer parameterization*, which applies passage knowledge injection only to a selected critical layer, greatly reducing injection cost. Experimental results on multi-hop QA and reasoning-aware knowledge editing showed that MERGEPRAG consistently outperforms standard and state-of-the-art RAG systems, existing PRAG methods, and fine-tuning–based parametric adaptation.

For future work, we plan to extend the framework to a more general reasoning-enhanced RAG setting to examine whether passage injection also contributes to further performance improvements. We also aim to explore the "pretraining" of hypernetworks, enabling them to be applied and adapted efficiently to new domains without requiring substantial additional training. Finally, we will investigate in depth why standard fine-tuning suffers from stronger performance degradation, whereas hypernetwork-based parameterization is helpful in boosting the performance. It is also worth exploring alternative hypernetwork architectures, such as memory-augmented designs, which can parameterize longer contexts more effectively beyond the single-passage setting used in this work.

### ACKNOWLEDGMENTS

This work was supported by Institute of Information & communications Technology Planning & Evaluation(IITP) grant funded by the Korea government(MSIT)(No.RS-2020-II201336, Artificial Intelligence graduate school support(UNIST)) and IITP grant funded by the Korea government(MSIT) (No.RS-2023-00216011, Development of artificial complex intelligence for conceptu-

ally understanding and inferring like human). Xuebing Liu and Shanbao Qiao were also supported by China Scholarship Council (CSC).

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

## A  LAYER SCANNING EXPERIMENTS FOR CRITICAL LAYER PARAMETERIZATION

The *critical-layer* parameterization module applies $\mathcal{H}$ only to a single critical layer $l^*$. To identify $l^*$, we perform a layer-wise scanning experiment that evaluates perplexity after adding a layer-specific paragraph vector to the $l$-th layer. For this purpose, we construct a small sub-dataset from the dataset used in the experiment.

Formally, let $\mathcal{H}_\psi^l$ denote the layer-specific hypernetwork for the $l$-th layer, parameterized by $\psi$ and defined following Eqs. 12–13. Given a question $q$, we first retrieve relevant passages $P \subseteq \mathcal{R}$. Each passage $p \in P$ is fed into $\mathcal{H}_\psi^l$ to obtain its passage expert $E_{\mathcal{H}_\psi(p)}$, which are then merged into a single expert $E_{\mathcal{H}_\psi}(P)$ using the inner-merging operation in Eq. 6. The merged expert is subsequently incorporated into the $l$-th layer of the base LLM $\mathcal{M}_{\theta_0}$ via Eq. 9. We train $\mathcal{H}_\psi^l$ by minimizing the cross-entropy loss defined in Eq. 14.

To measure the importance of each layer $l$, we evaluate perplexity after training $\mathcal{H}_\psi^l$. Figures 2a, 2b, 2c, 3a, 3b and 3c compare perplexity across layers for LLaMA3.1-8B and Qwen2.5-7B on different experimental dataset, respectively, under the setting of $|SP| = 1$.

The results show a clear sensitivity pattern: in LLaMA3.1-8B and Qwen2.5-7B, injecting passage vectors into early-to-middle layers yields the largest perplexity reduction, indicating that these layers play a central role in integrating external knowledge. Meanwhile, we observe that the two models exhibit opposite patterns in the layers where external knowledge is least efficiently integrated. Specifically, LLaMA3.1-8B shows higher perplexity when the passage vector is injected into the shallowest layers, whereas Qwen2.5-7B displays higher perplexity when the injection is applied to the deepest layers. This contrast suggests that conducting layer-wise scanning is essential for identifying the optimal injection layer for different model architectures.

Overall, our findings show that both LLaMA3.1-8B and Qwen2.5-7B exhibit their highest sensitivity to passage-vector injection in the early-to-middle layers, suggesting that these layers are primarily responsible for incorporating external knowledge across model families. Based on the layer-wise perplexity analysis conducted on three datasets—HotpotQA, WikiMhQA, and Musique—we select the optimal insertion layer $l^*$ for each model–dataset pair. The selected layers are summarized in Table 8.

| Model | HotpotQA | 2WikiMultihopQA | MuSiQue |
|---|---|---|---|
| LLaMA3.1-8B | $l^* = 9$ | $l^* = 7$ | $l^* = 8$ |
| Qwen2.5-7B | $l^* = 7$ | $l^* = 8$ | $l^* = 9$ |

Table 8: Selected critical layers $l^*$ for passage-vector insertion based on layer-wise perplexity analysis across datasets.

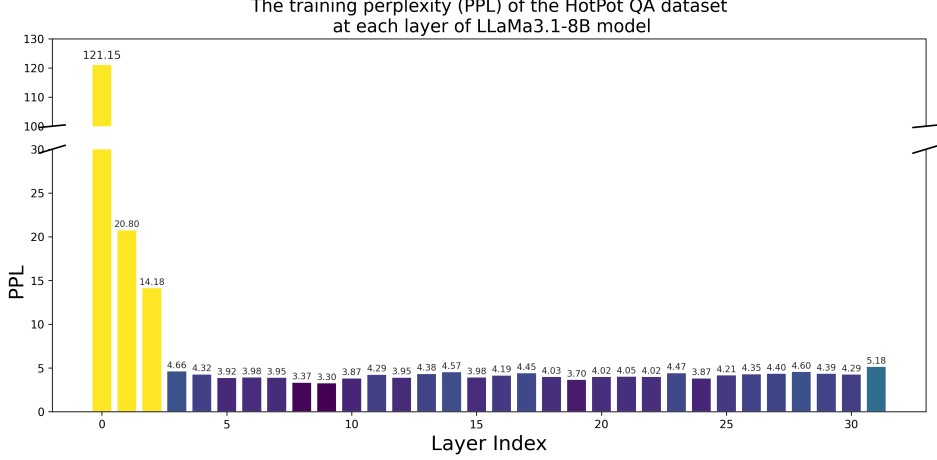

(a) HotpotQA: layer-wise perplexity ($|SP| = 1$)

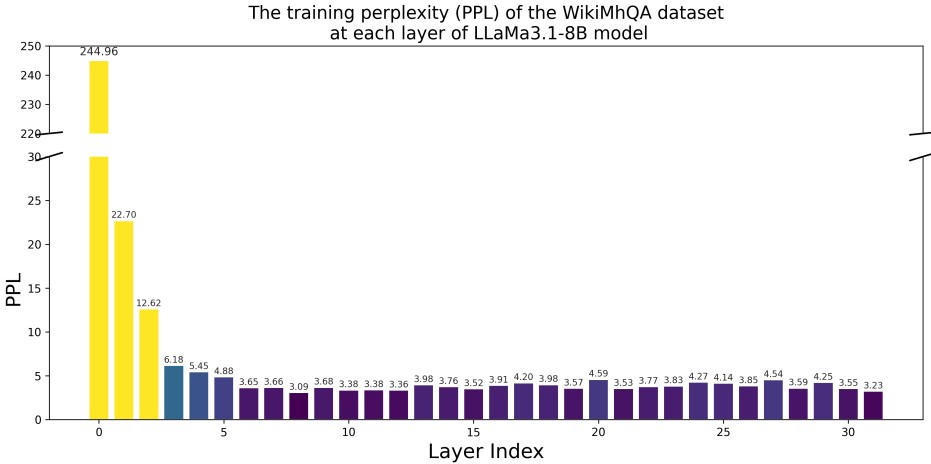

(b) 2WikiMhQA: layer-wise perplexity ($|SP| = 1$)

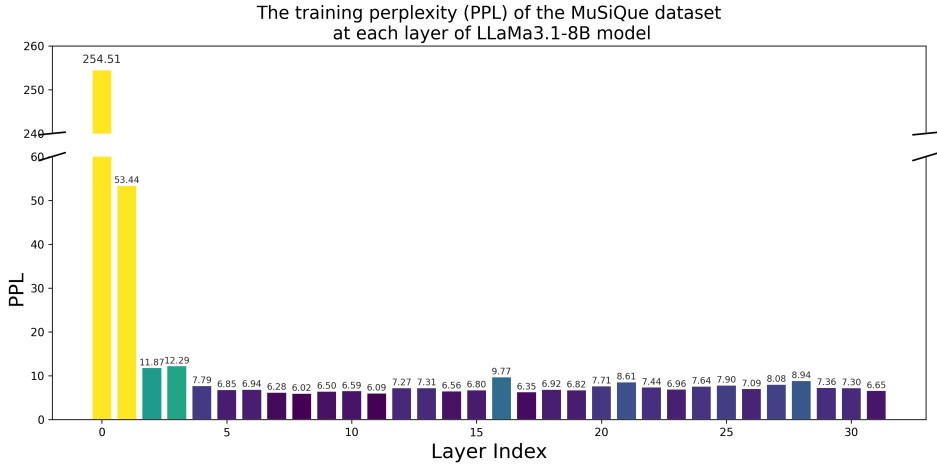

(c) MuSiQue: layer-wise perplexity ($|SP| = 1$)

Figure 2: Layer-wise perplexity trends of LLaMA3.1-8B with paragraph-vector insertion on Hot-potQA, 2WikiMhQA, and MuSiQue under $|SP| = 1$.

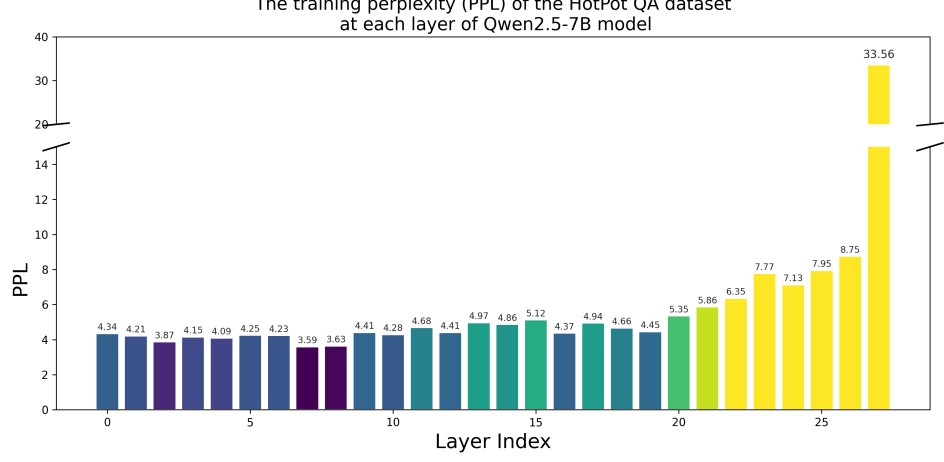

(a) HotpotQA: layer-wise perplexity ($|SP| = 1$)

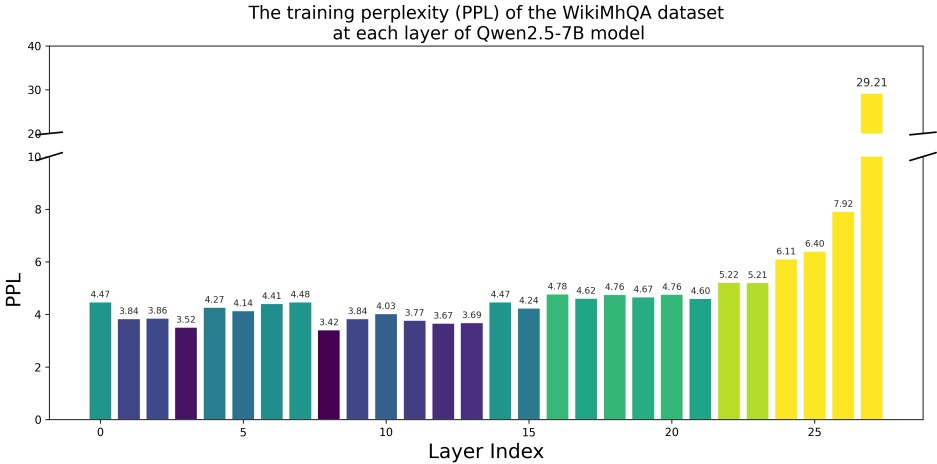

(b) 2WikiMhQA: layer-wise perplexity ($|SP| = 1$)

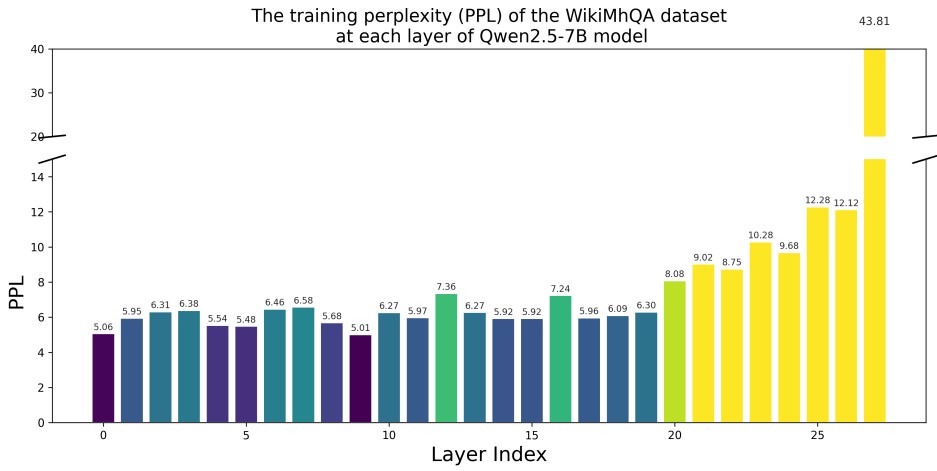

(c) MuSiQue: layer-wise perplexity ($|SP| = 1$)

Figure 3: Layer-wise perplexity trends of Qwen2.5-7B with paragraph-vector insertion on HotpotQA, 2WikiMhQA, and MuSiQue under $|SP| = 1$.

## B  ORTHOGONAL MERGING USING THE GRAM-SCHMIDT PROCEDURE

In multi-hop RAG, a set of passages $SP_i$ arrives for each sub-question $sq_i$. This setting naturally motivates the design of a *continual merging* mechanism that combines previously accumulated knowledge with newly retrieved passage knowledge, recurrently updating the current FFN expert by incorporating each new expert.

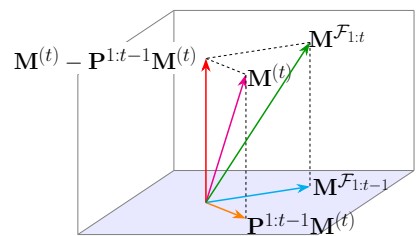

Figure 4: Illustration of orthogonal continual merging based on Gram–Schmidt procedure.

To minimize overwriting previously acquired knowledge, MergePRAG adopts *orthogonal continual merging* based on the Gram–Schmidt process, inspired by recent orthogonal approaches in model merging and knowledge editing (Xu et al., 2025). Specifically, the new parameter matrix is projected onto the span of the previously merged parameters, and only its orthogonal residual is added to the current merged expert.

We apply orthogonal continual merging separately to either key or value matrices, resulting from $\mathcal{H}_\phi$. Formally, let $\{\mathbf{M}^{(i)}\}_{i=1}^{t}$ denote the sequence of key or value passage memories, where each $\mathbf{M}^{(i)} \in \mathcal{H}_\phi(SP_i)$ corresponds to either $\mathbf{K}_p$ or $\mathbf{V}_p$.

Suppose that $\mathbf{M}^{\mathcal{F}_{1:t-1}}$ denotes the merged memory parameter obtained from $\{\mathbf{M}^{(i)}\}_{i=1}^{t-1}$. Following Eq. 10 in Section 3.2.2, the Gram–Schmidt orthogonalization procedure first computes the projection matrix onto the subspace spanned by $\mathbf{M}^{\mathcal{F}_{1:t-1}}$:

$$\mathbf{P}^{1:t-1} = \mathbf{M}^{\mathcal{F}_{1:t-1}}\big((\mathbf{M}^{\mathcal{F}_{1:t-1}})^\top \mathbf{M}^{\mathcal{F}_{1:t-1}}\big)^{-1}(\mathbf{M}^{\mathcal{F}_{1:t-1}})^\top. \tag{15}$$

The new parameter $\mathbf{M}^{(t)}$ is then merged by adding only its orthogonal component with respect to the subspace spanned by $\mathbf{M}^{\mathcal{F}_{1:t-1}}$:

$$\mathbf{M}^{\mathcal{F}_{1:t}} = \mathbf{M}^{\mathcal{F}_{1:t-1}} + \big(\mathbf{I} - \mathbf{P}^{1:t-1}\big)\mathbf{M}^{(t)}, \tag{16}$$

where $\mathbf{P}^{1:t-1}$ is the projection matrix defined in Eq. 15.

With a slight abuse of notation, the recursion in Eq. 16 is denoted by Merge:

$$\mathbf{M}^{\mathcal{F}_{1:t}} = \mathsf{Merge}\Big(\mathbf{M}^{\mathcal{F}_{1:t-1}}, \mathbf{M}^{(t)}\Big). \tag{17}$$

When the merging procedure Merge is applied independently to the sequences of key and value matrices $\mathbf{K}_p$ and $\mathbf{V}_p$, we obtain the merged passage memories for both parts:

$$\begin{aligned}
\mathbf{K}^{\mathcal{F}_{1:t}} &= \mathsf{Merge}\Big(\mathbf{K}^{\mathcal{F}_{1:t-1}}, \mathbf{K}^{(t)}\Big), \\
\mathbf{V}^{\mathcal{F}_{1:t}} &= \mathsf{Merge}\Big(\mathbf{V}^{\mathcal{F}_{1:t-1}}, \mathbf{V}^{(t)}\Big).
\end{aligned} \tag{18}$$

where $\mathbf{K}^{(t)}$ and $\mathbf{V}^{(t)}$ denote the key and value memory matrices at timestep $t$, respectively. Here, Merge refers to either $\mathsf{Merge}_{\mathrm{seq}}$ in Eq. 3 or $\mathsf{Merge}_{\mathrm{inner}}$ in Eq. 6.

## C  HYPERNETWORK ARCHITECTURE

As shown in Figure 5, given a passage $p$, the hypernetwork $\mathcal{H}_\phi$ generates the corresponding key–value memory through three stages: (1) *Attentive pooling*, which produces a sentence-level embedding $\mathsf{Emd}(p)$ for passage $p$; (2) *MLP*, which maps the passage embedding to a latent representation using a two-layer ReLU-based MLP; and (3) *Linear projection*, which converts the latent representation into key–value vectors $\mathbf{K}_p, \mathbf{V}_p$, yielding the passage-specific memory.

We define a lightweight Encoder consisting of a 2-layer Transformer encoder layer with 4 attention head. The hidden dimension of the encoder layer is set to be consistent with the LM model's internal representation dimension $d$. Specifically, for LLaMA3.1-8B, $d = 4096$, while for Qwen2.5-7B, $d = 3584$.

Figure 5: The hypernetwork $\mathcal{H}_\phi(p)$ generates passage-specific key–value vectors $\mathbf{K}_p, \mathbf{V}_p \in \mathbb{R}^{k \times d_{\text{model}}}$, referred to as passage-specific memory, which serve as lightweight, plug-in passage-level experts for downstream reasoning. The process consists of three stages: (1) **attentive pooling**, (2) **MLP**, and (3) **linear projection**. *1) Attentive pooling.* Given a one-hot token matrix $\mathbf{X} \in \mathbb{R}^{T \times |\mathcal{V}|}$ for passage $p$, the model first converts it into a sequence of embeddings $\mathbf{X}_{\text{emb}} \in \mathbb{R}^{T \times d}$ via the word embedding layer: $\mathbf{X}_{\text{emb}} = \mathsf{Embedding}(\mathbf{X})$ (Eq. 19). Attention is then applied over the token embeddings, where a learnable vector $\mathbf{w}_a \in \mathbb{R}^d$ serves as the query: $\mathsf{Emd}(p) = \mathbf{h} \in \mathbb{R}^d$ (Eq. 20). *2) MLP.* The pooled representation $\mathbf{h}$ is passed through a two-layer feedforward network with ReLU activations and LayerNorm, producing a latent representation $\mathbf{h}_b$ (Eq. 21). *3) Linear projection.* Two independent linear projection heads map $\mathbf{h}_b$ into the key and value parameter spaces: $\mathbf{K}_p, \mathbf{V}_p \in \mathbb{R}^{k \times d_{\text{model}}}$, yielding flattened key–value memory vectors of length $k \cdot d_{\text{model}}$ for passage $p$ (Eq. 22). The resulting passage-specific memory is subsequently injected into the target model as additional knowledge signals.

**Attentive pooling.** Given a passage $p$, we apply attention-based aggregation over token-level embeddings of $p$: (1) obtaining sequence of its word embeddings, and (2) applying attentive pooling. Formally, let the retrieved passage be represented as a sequence of tokens, denoted by $\mathbf{X} \in \mathbb{R}^{T \times |\mathcal{V}|}$, where each row $\mathbf{X}_t$ is a one-hot vector over the vocabulary indicating the identity of the token at the $t$-th position, and $\mathcal{V}$ denotes the vocabulary set. We apply word embedding layer $\mathsf{Embedding}$ to $\mathbf{X}$ and obtain its embedded representations, as follows:

$$\mathbf{X}_{emd} = \mathsf{Embedding}(\mathbf{X}). \tag{19}$$

where a sequence of token embeddings $\mathbf{X}_{emd} \in \mathbb{R}^{T \times d}$, where $T$ is the passage length and $d$ is the embedding dimension. Note that word embedding layer $\mathsf{Embedding}$ is obtained from the pretrained LLM $\mathcal{M}_{\theta_0}$ (e.g., LLaMA3.1-8B or Qwen2.5-7B).

The passage embedding $\mathsf{Emd}(p)$ is then obtained via attentive pooling:

$$\mathsf{Emd}(p) \quad = \quad \mathbf{h} = \mathrm{softmax}\big(\mathbf{w}_a^\top \mathbf{X}_{emd}^\top\big) \mathbf{X}_{emd} \in \mathbb{R}^d, \tag{20}$$

where $\mathbf{w}_a \in \mathbb{R}^d$ is a learnable attention vector[4]. The embedding $\mathsf{Emd}(p) = \mathbf{h}$ serves as the attentively pooled representation of the passage, capturing its global semantic content.

**MLP.** To increase representational capacity and allow the hypernetwork to perform nonlinear reasoning over the passage summary, the pooled vector $\mathsf{Emd}(p) = \mathbf{h}$ is passed through a two-layer feedforward network, denoted $\mathsf{MLP}_{\text{hyp}}$, as follows:

$$\mathbf{h}_b = \mathsf{MLP}_{\text{hyp}}(\mathbf{h}) = \mathrm{ReLU}\big(\mathbf{V}' \, \mathrm{LN}\left(\mathrm{ReLU}(\mathbf{W}'\mathbf{h})\right)\big). \tag{21}$$

where LN is the layer normalization layer.

---

[4]We omit an additional bias term as it has negligible impact.

**Linear projection.**  Finally, two linear transformations map the latent code $\mathbf{h}_b$ into flattened key and value matrices, i.e., the passage-specific memory:

$$\begin{aligned}
\mathbf{K}_p &= \mathbf{W}_K\,\mathbf{h}_b + \mathbf{b}_K, \\
\mathbf{V}_p &= \mathbf{W}_V\,\mathbf{h}_b + \mathbf{b}_V,
\end{aligned} \tag{22}$$

where $k$ (i.e., *num-kv*) denotes the number of key–value slots generated per passage and $d$ is the model dimension. Each of the $k$ rows corresponds to an independent memory vector that can be directly attended to by the language model.

This three-stage design enables the hypernetwork to compress an entire retrieved passage into a compact set of attention-ready memory vectors, which are efficiently integrated into the model via the memory attention mechanism at the designated target layer (Eq. 8 in Section 3.2.1).

## D  TRAINING AND INFERENCE PROCEDURE

### D.1  TRAINING

**Hypernetwork.**  We prepare a training dataset $\mathcal{D}_\mathcal{H}$ at the sub-question level from training set in each task to train the hypernetwork $\mathcal{H}_\phi$. Each instance $(q, p, a) \in \mathcal{D}_\mathcal{H}$ consists of a sub-question $q$, its gold passage $p$, and the corresponding answer $a$. The hypernetwork parameters $\phi$ are trained by minimizing the negative log-likelihood of generating the correct answer $a$ under $\mathcal{M}_{\theta_0 \oplus \mathcal{H}(p)}$ (Eq. 14), while the base parameters $\theta_0$ remain frozen.

**Subquestion generator.**  To train the sub-question generator $\mathcal{M}_{sq}$, we construct a dataset $\mathcal{D}_{sq} = \{(q^{(j)}, y^{(j)})\}_{j=1}^{M}$ using GPT-4.1 (Achiam et al., 2023) to generate sub-questions from 4,000 randomly sampled examples in the training split of each dataset. The prompt template used for this dataset construction is shown in Table 9. The template specifies the desired output format and includes several illustrative examples. Given an input question $q$ and its associated gold passages, GPT-4.1 refers to the examples and decomposes $q$ into sub-questions using the provided passages.

Each target sequence is $y^{(j)} = [\,sq_1^{(j)}, sa_1^{(j)}, sq_2^{(j)}, sa_2^{(j)}, \ldots, sq_{n_j}^{(j)}, sa_{n_j}^{(j)}, \langle \mathrm{EOS} \rangle\,]$, as described in Section 4.1.

Given a pair $(q^{(j)}, y^{(j)})$, $\mathcal{M}_{sq}$ is trained with supervised fine-tuning by minimizing the negative log-likelihood (NLL):

$$\mathcal{L}(\mathcal{M}_{sq}) = -\sum_{j=1}^{M}\sum_{t=1}^{|y^{(j)}|} \log P_{\mathcal{M}_{sq}}\left(y_t^{(j)} \mid q^{(j)}, y_{<t}^{(j)}\right), \tag{23}$$

where $y_t^{(j)}$ denotes the $t$-th token in the target sequence $y^{(j)}$.

### D.2  INFERENCE

At inference time, MergePRAG tackles a multi-hop QA task by decomposing the original complex question into sub-questions. For each sub-question $sq_i$, the top-retrieved passages $SP_i$ are fed into the hypernetwork $\mathcal{H}_\phi$ to produce a sub-expert $E_{\mathcal{H}_\phi}(SP_i)$. This sub-expert is then merged with the previously accumulated FFN expert $E_\mathcal{F}(SP_{1:i-1})$ using orthogonal continual merging, yielding the updated fused expert $E_\mathcal{F}(SP_{1:i})$, ensuring that knowledge from earlier reasoning steps is preserved without redundancy.

The fused FFN expert is injected into the base LLM $\mathcal{M}_{\theta_0}$ at the critical layer $l^*$. Response generation is then performed under the updated model $\mathcal{M}_{\theta_0 \oplus \mathcal{F}(SP_{1:i})}$, either with the current in-context passages $SP_i$ (MergePRAG+) or without them (MergePRAG). After all sub-questions are processed at timestep $T$, the final answer to the original complex question is generated by the fully passage-injected model $\mathcal{M}_{\theta_0 \oplus \mathcal{F}(SP_{1:T})}$. The complete inference procedure is summarized in Algorithm 1. For comparison, the inference procedure of MultihopRAG without passage knowledge parameterization is shown in Algorithm 2.

Your task is to convert a given question and its related facts into a multi-step reasoning chain.

Requirements: Each step in the reasoning chain must:
- Use one fact from the input facts, do not combine, summarize, or fabricate facts; each fact must be used as-is from the input.
- Generate a "Sub-question" and a short answer "Sub-answer".
- The answer "Sub-answer" must be directly derivable from the corresponding Fact.

Examples:
Question: "When did the civilisation start that Desalpar Gunthli was a part of?",
Facts: ["Desalpar Gunthli: Desalpar Gunthli is a village and site belonging to Indus Valley Civilisation located at Nakhtrana Taluka, Kutch District, Gujarat, India.", "Indus Valley Civilisation: The Indus Valley Civilisation (IVC) or Harappan Civilisation was a Bronze Age civilisation (3300–1300 BCE; mature period 2600–1900 BCE) mainly in the northwestern regions of South Asia, extending from what today is northeast Afghanistan to Pakistan and northwest India."]
Output: [
{
"Sub-question": "Which civilisation was Desalpar Gunthli a part of?",
"Fact": "Desalpar Gunthli: Desalpar Gunthli is a village and site belonging to Indus Valley Civilisation located at Nakhtrana Taluka, Kutch District, Gujarat, India.",
"Sub-answer": "Indus Valley Civilisation"
},
{
"Sub-question": "When did the Indus Valley Civilisation exist?",
"Fact": "Indus Valley Civilisation: The Indus Valley Civilisation (IVC) or Harappan Civilisation was a Bronze Age civilisation (3300–1300 BCE; mature period 2600–1900 BCE) mainly in the northwestern regions of South Asia, extending from what today is northeast Afghanistan to Pakistan and northwest India.",
"Sub-answer": "3300–1300 BCE"
} ]

**[other examples demonstrations abbreviated]**

Question: {}
Facts: {}
Output:

Table 9: Prompt templates used by GPT-4.1 to generate sub-questions and sub-answers from the training set. The generated results are used to construct $\mathcal{D}_{sq}$ for training the sub-question generator.

# E  FURTHER EXPERIMENT RESULTS

## E.1  MULTI-HOP EDITING

The MQuAKE (Zhong et al., 2023) benchmark consists of two multi-hop editing datasets: MQuAKE-CF and MQuAKE-T. MQuAKE-CF is a counterfactual knowledge editing dataset designed to evaluate how effectively models can incorporate and generalize from counterfactual modifications to existing knowledge. In contrast, MQuAKE-T targets temporal knowledge updates, assessing a model's ability to track and apply changes in real-world facts over time.

As shown in Table 10, MergePRAG achieves superior performance compared with prompt-based methods such as RAG and MeLLo, which rely on explicit prompting. These results further demonstrate the effectiveness of injecting knowledge into the model for multi-hop editing tasks as well.

## E.2  EFFICIENCY ANALYSIS

To evaluate the inference cost of MergePRAG, we conduct an efficiency analysis (Table 11) focusing on three components of the system: (1) passage-specific memory generation by $\mathcal{H}_\psi$, (2) sub-question

---

**Algorithm 1** Multi-hop Inference with MergePRAG

---

**Require:** Original question $q$, sub-question generator $\mathcal{M}_{sq}$, base LLM $\mathcal{M}_{\theta_0}$, hypernetwork $\mathcal{H}_\phi$, retriever $\mathcal{R}$
**Ensure:** Final answer $a$
 1: Initialize merged expert $\mathcal{F} \leftarrow \emptyset$
 2: Initialize reasoning chain $\mathcal{C} \leftarrow \emptyset$
 3: **while** next sub-question exists **do**
 4:     Generate sub-question: $sq_i \leftarrow \mathcal{M}_{sq}(q, \mathcal{C})$
 5:     Retrieve passages: $SP_i \leftarrow \mathcal{R}(sq_i)$
 6:     Parameterize passages: $\mathcal{H}_\phi(SP_i) \leftarrow \text{Merge}_{\text{inner}}(\{\mathcal{H}_\phi(p) \mid p \in SP_i\})$           (Eq. 6)
 7:     **if** $i > 1$ **then**
 8:         Orthogonal continual merge: $\mathcal{F} \leftarrow \text{Merge}_{seq}(\mathcal{F}, \mathcal{H}_\phi(SP_i))$           (Sec. 3.2.2)
 9:     **else**
10:         Initialize expert: $\mathcal{F} \leftarrow \mathcal{H}_\phi(SP_i)$
11:     **end if**
12:     Inject $\mathcal{F}$ into the base LLM: $\mathcal{M}_{\theta_0 \oplus \mathcal{F}}$
13:     Generate sub-answer:

$$sa_i \leftarrow \begin{cases} \mathcal{M}_{\theta_0 \oplus \mathcal{F}}(sq_i), & \text{(MergePRAG)} \\ \mathcal{M}_{\theta_0 \oplus \mathcal{F}}(SP_i, sq_i), & \text{(MergePRAG+)} \end{cases}$$

14:     Append $(sq_i, sa_i)$ to reasoning chain $\mathcal{C}$
15: **end while**
16: Generate final answer:

$$a \leftarrow \mathcal{M}_{\theta_0 \oplus \mathcal{F}}(\mathcal{C}, q) \quad \text{(MergePRAG / MergePRAG+)}$$

17: **return** $a$

---

**Algorithm 2** Multi-hop Inference with MultihopRAG

---

**Require:** Original question $q$, sub-question generator $\mathcal{M}_{sq}$, base LLM $\mathcal{M}_{\theta_0}$, retriever $\mathcal{R}$
**Ensure:** Final answer $a \leftarrow \emptyset$
 1: Initialize reasoning chain $\mathcal{C} \leftarrow \emptyset$
 2: **while** next sub-question exists **do**
 3:     Generate sub-question: $sq_i \leftarrow \mathcal{M}_{sq}(q, \mathcal{C})$
 4:     Retrieve passages: $SP_i \leftarrow \mathcal{R}(sq_i)$
 5:     Generate sub-answer: $sa_i \leftarrow \mathcal{M}_{\theta_0}(SP_i, sq_i)$
 6:     Append $(sq_i, sa_i)$ to reasoning chain $\mathcal{C}$
 7: **end while**
 8: Generate final answer: $a \leftarrow \mathcal{M}_{\theta_0}(\mathcal{C}, q)$
 9: **return** $a$

---

response generation, and (3) overall response generation. Thanks to the lightweight design of the hypernetwork, the time required to produce passage-specific key–value memory is minimal. The subsequent step of generating sub-questions also incurs only modest overhead.

Although decomposing a complex query into multiple sub-questions increases the number of inference steps compared with standard RAG, the overall latency remains within a practical range. Notably, the proposed pipeline still requires less time than RAG-CoT methods, which rely on long chain-of-thought prompting, while achieving higher accuracy. These results demonstrate that MergePRAG offers a favorable trade-off between computational efficiency and reasoning effectiveness.

## F  BASELLINES INTRODUCTION

**RAG:** (Lewis et al., 2020) For a given query $q$, the retriever selects the top-$k$ relevant passages. The generator then directly infers the answer based on these retrieved passages. To ensure stylistic

| Model | Method | MQuAKE-CF | | MQuAKE-T | |
|---|---|---|---|---|---|
| | | EM | F1 | EM | F1 |
| | RAG | 4.48 | 9.27 | 27.69 | 31.92 |
| LLaMA3.1-8B | RAG-CoT | 11.7 | 13.18 | 45.93 | 47.28 |
| | MeLLo | 32.90 | 34.10 | 85.40 | 86.21 |
| | **MergePRAG+**$_{|SP|=1}$ | **50.30** | **51.36** | **96.10** | **96.10** |

Table 10: Results on the multi-hop editing task under the MQuAKE datasets.

| | Generate Passage Memory | Response $sa$ | Average Time |
|---|---|---|---|
| RAG$_{|SP|=1}$ | - | - | 0.712s |
| RAG-CoT$_{|SP|=1}$ | - | - | 6.389s |
| MergePRAG+$_{|SP|=1}$ | 0.001s | 0.259s | 2.517s |

Table 11: Efficiency analysis of MergePRAG using the LLaMA3.1-8B model on the HotpotQA dataset.

consistency of the generated answers, we apply a task-specific prompt. The RAG prompt template is provided in Table 13.

**RAG-CoT:** (Wei et al., 2022) Building upon RAG, RAG-CoT incorporates chain-of-thought reasoning. To guide the model's reasoning process, we employ a one-shot demonstration sampled from the training data, which encourages the model to generate step-by-step explanations before arriving at the final answer. The prompt template is included in Table 14. We evaluate RAG and RAG-CoT by retrieving 1–8 relevant passages, with the accuracy results shown in Figure 6.

**MulitihopRAG:** MultihopRAG can be viewed as a variant of MergePRAG without $\mathcal{H}$, which iteratively responds to sub-questions using a pure RAG-style approach (Algorithm 2).

**IRCoT:** (Trivedi et al., 2023) Interleaves retrieval with chain-of-thought reasoning, enabling iterative evidence retrieval conditioned on intermediate reasoning steps, which enhances multi-hop QA performance and reduces hallucination.

**MeLLo:** (Zhong et al., 2023) MeLLo is a system that iteratively decomposes multi-hop questions into subquestions, generates tentative answers, retrieves relevant facts, and updates predictions based on potential contradictions.

**FLARE:** (Jiang et al., 2023) Incorporates adaptive retrieval triggered when the model generates low-confidence tokens, leveraging retrieved evidence to improve response quality.

**Adaptive-RAG:** (Jeong et al., 2024) Adaptive-RAG automatically selects the optimal retrieval and reasoning strategy based on query complexity, ensuring efficient handling of simple queries while improving accuracy on complex ones.

**Auto-RAG:** (Yu et al., 2024b) It performs iterative reasoning to decide when and what to retrieve, and terminates the process once sufficient external knowledge has been gathered, before generating the final answer.

**DeepRAG:** (Guan et al., 2025) It odels retrieval-augmented generation as a Markov decision process, where the query is iteratively decomposed and the model dynamically decides at each step whether to retrieve external knowledge or rely on parametric reasoning.

**R3-RAG:** (Li et al., 2025b) It is a reinforcement learning–based method that trains LLMs to iteratively reason and retrieve, enabling them to acquire more comprehensive external knowledge and generate more accurate answers.

**Search-o1:** (Li et al., 2025a) Search-o1 lets a reasoning model dynamically retrieve and analyze external knowledge to fill knowledge gaps during long reasoning.

**Search-R1:** (Jin et al., 2025) Search-R1 enables an LLM to learn, via reinforcement learning, how to autonomously issue effective multi-turn search queries during step-by-step reasoning, thereby substantially improving retrieval-augmented QA performance.

| Model | Method | Retriever | HotpotQA | | 2WikiMhQA | | MuSiQue | |
|---|---|---|---|---|---|---|---|---|
| | | | EM | F1 | EM | F1 | EM | F1 |
| FLAN-T5-XL | Adaptive-RAG | BM25 | 42.00 | 53.82 | 40.60 | 49.75 | 23.60 | 31.80 |
| LLaMA3.1-8B | Search-o1 | BM25 | 14.80 | 24.08 | 22.20 | 27.10 | 5.40 | 11.98 |
| Qwen2.5-7B | Search-o1 | BM25 | 11.60 | 16.95 | 22.00 | 25.02 | 2.10 | 7.48 |
| Qwen2.5-3B | Search-R1 | E5 | 32.40 | - | 31.90 | - | 10.30 | - |
| Qwen2.5-7B | Search-R1 | E5 | 37.00 | - | 41.40 | - | 14.60 | - |
| Qwen2.5-7B | MergePRAG$_{|SP|=1}$ | E5 | 43.40 | 50.64 | 65.80 | 69.72 | 9.70 | 19.61 |
| Qwen2.5-7B | MergePRAG$_{|SP|=1}$ | BM25 | 42.00 | 49.09 | 59.70 | 63.05 | 13.00 | 23.35 |
| LLaMA3.1-8B | MergePRAG$_{|SP|=1}$ | E5 | 48.80 | 55.53 | 66.30 | 71.05 | 14.40 | 25.04 |
| LLaMA3.1-8B | MergePRAG$_{|SP|=1}$ | BM25 | 46.80 | 53.40 | 61.60 | 67.31 | 17.80 | 29.39 |

Table 12: Performance comparison of MergePRAG+ with other advanced RAG methods on three QA benchmarks – Adaptive-RAG, Search-R1 and Search-o1.

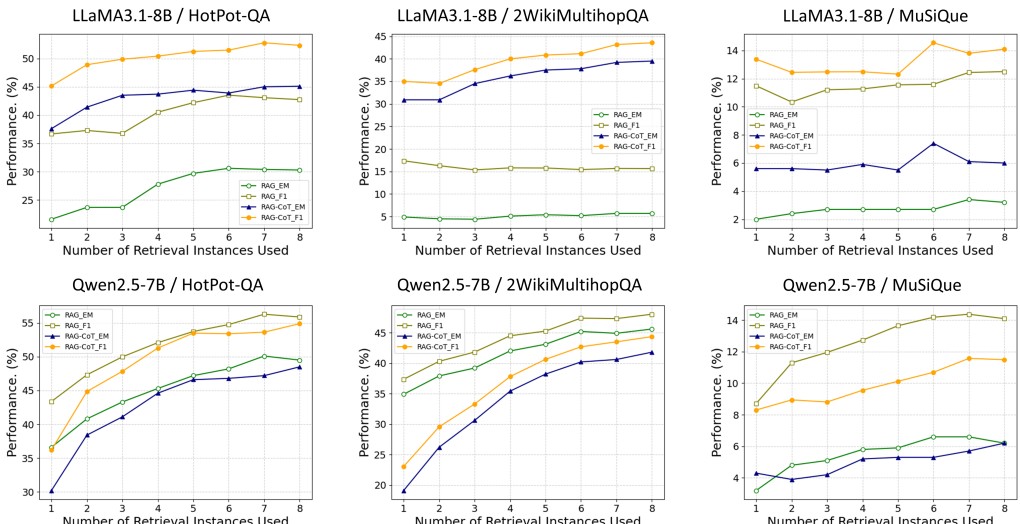

Figure 6: Results of RAG and RAG-CoT varying the number of retrieved passages on three multi-hop QA datasets using LLaMA3.1-8B and Qwen2.5-7B.

**PRAG+:** (Su et al., 2025) By transforming the documents retrieved for query $q$ into parametric representations that are directly integrated into the feed-forward networks of the LLM, parametric retrieval-augmented generation is introduced.

**DyPRAG+:** (Tan et al., 2025a) Extends PRAG by employing a lightweight parameter transformation module to efficiently convert documents retrieved for query $q$ into parametric knowledge, which can be directly leveraged to generate the response.

## G  MERGING METHODS: INTRODUCTION

**Arithmetic mean merging.** Arithmetic mean merging computes the element-wise mean of the task vectors $\{\boldsymbol{\tau}_j\}_{j=1}^n$, where $n$ is the number of tasks. This approach assumes that all vectors lie in a shared embedding space and produces a balanced fusion without introducing additional learnable parameters:

$$\mathsf{Merge}\big(\{\boldsymbol{\tau}_j\}_{j=1}^n\big) = \frac{1}{n}\sum_{j=1}^n \boldsymbol{\tau}_j. \tag{24}$$

**Additive merging.** Additive merging performs element-wise summation of task vectors. This operation preserves activation magnitudes and emphasizes consistently high-valued features across task

Follow the format below to answer the following question with a very short phrase, such as "1998", "May 16th, 1931", or "James Bond", to meet the criteria of exact match datasets.

Passage: {}
Question: {}
Answer:

Table 13: Input template used for evaluating multi-hop questions with RAG.

You are a reasoning assistant tasked with answering user questions step by step. Follow the format below to answer the following question with a very short phrase, such as "1998", "May 16th, 1931", or "James Bond", to meet the criteria of exact match datasets.

**Passage:** Tom Warburton: Since moving to Los Angeles in 2009 he has worked at Disney Television Animation serving as creative director on "Fish Hooks" and co-executive producer on "The 7D". Fish Hooks: Fish Hooks is an American animated television series created by Noah Z. Jones that originally aired on Disney Channel from September 3, 2010 to April 4, 2014.
**Question:** What show that Tom Warburton worked on aired from September 3, 2010 to April 4, 2014?
**Thoughts**: The passage says Tom Warburton worked as creative director on Fish Hooks. The passage also says Fish Hooks aired on Disney Channel from September 3, 2010 to April 4, 2014. The question asks which show that Tom Warburton worked on aired during those dates. So, the answer must be Fish Hooks.
**Answer:** Fish Hook

Passage: {}
Question: {}
Thoughts:

Table 14: Input template used for evaluating multi-hop questions with RAG-CoT.

vectors, without trainable parameters, as follows:

$$\mathsf{Merge}\big(\{\boldsymbol{\tau}_j\}_{j=1}^n\big) = \sum_{j=1}^n \boldsymbol{\tau}_j. \tag{25}$$

**Concat merging.** Concat merging first concatenates the task vectors and then applies a learnable linear projection to map the concatenated vector into the merged space:

$$\mathsf{Merge}\big(\{\boldsymbol{\tau}_j\}_{j=1}^n\big) = \mathrm{concat}\left(\boldsymbol{\tau}_1, \ldots, \boldsymbol{\tau}_n\right). \tag{26}$$

In our setting, concatenation is equivalent to increasing the number of key–value vectors from $k$ to $k \times n$. For example, suppose $\mathbf{K}^{(1)}, \mathbf{K}^{(2)} \in \mathbb{R}^{k \times d_{\mathrm{model}}}$ are the key memories from two tasks. Concat merging produces:

$$\mathsf{Merge}\Big(\mathbf{K}^{(1)}, \mathbf{K}^{(2)}\Big) = \mathrm{concat}\Big(\mathbf{K}^{(1)}, \mathbf{K}^{(2)}\Big) \in \mathbb{R}^{2k \times d_{\mathrm{model}}}. \tag{27}$$

**TIES merging.** TIES merging (Yadav et al., 2023) (Trim–Elect–Sign Merging) fuses task vectors by retaining only the largest-magnitude and sign-consistent components across tasks. This approach preserves salient and mutually aligned activations while suppressing contradictory or noisy features. Given $n$ task vectors, TIES merging proceeds in three stages:

**1. Trim.** Given a task vector $\boldsymbol{\tau}_j$, the trimming step applies magnitude-based pruning:

$$\hat{\boldsymbol{\tau}}_j = \mathrm{top}_k(\boldsymbol{\tau}_j), \tag{28}$$

where $\mathrm{top}_k$ retains the top $k\%$ of parameters by magnitude and sets the remaining entries to zero. The trimmed vector is decomposed into its sign and magnitude components:

$$\hat{\boldsymbol{\tau}}_j = \hat{\boldsymbol{\gamma}}_j \odot \hat{\boldsymbol{m}}_j, \tag{29}$$

where

$$\hat{\boldsymbol{\gamma}}_j = \text{sign}(\hat{\boldsymbol{\tau}}_j), \qquad \hat{\boldsymbol{m}}_j = |\hat{\boldsymbol{\tau}}_j|,$$

and $\odot$ denotes element-wise multiplication.

**2. Elect.** The election step performs magnitude-weighted sign aggregation. The merged sign vector is computed by selecting, for each coordinate, the sign with the largest summed magnitude across all trimmed task vectors:

$$\boldsymbol{\gamma}_m = \text{sign}\left(\sum_{j=1}^{n} \hat{\boldsymbol{\tau}}_j\right). \tag{30}$$

**3. Merge.** Given the trimmed task vectors $\hat{\boldsymbol{\tau}}_j$, the merging step selectively aggregates only those coordinates whose signs match the elected sign $\boldsymbol{\gamma}_m$. Formally,

$$\begin{aligned}
\boldsymbol{a}_j &= \mathcal{I}(\hat{\boldsymbol{\gamma}}_j = \boldsymbol{\gamma}_m), \\
\boldsymbol{\tau}_m &= \left(\sum_{j=1}^{n} \hat{\boldsymbol{\tau}}_j \odot \boldsymbol{a}_j\right) \oslash \left(\sum_{j=1}^{n} \boldsymbol{a}_j\right),
\end{aligned} \tag{31}$$

where $\mathcal{I}(e)$ is the indicator function that returns 1 if the condition $e$ is true and 0 otherwise, and $\oslash$ denotes element-wise division.

## H  CASE STUDY

We present a case study to illustrate the decomposition process. As shown in Table 16, the sub-question generator iteratively breaks down the question into sub-questions. For each step in Table 16, the upper part above the dashed line corresponds to the input template used by the sub-question generator, while the lower part shows the retrieval and sub-answer generation process. Green text denotes retrieved content, and red text indicates generated sub-answers. When the sub-question generator produces no further sub-questions, the resulting chain $C$ is passed into the merged-expert LM model. The inference process is illustrated in Table 15, where blue text highlights the final answer.

We further provide an error-case analysis, as shown in Table 17. This failure is triggered by an incorrect retrieval result for one of the sub-questions, which leads to an erroneous sub-answer. The mistake then propagates to subsequent steps, causing the next sub-question to deviate from the original problem and ultimately resulting in a chain reaction of compounding errors. As indicated by our ablation study on the number of retrieved documents (i.e., cases where $|SP| > 1$), increasing retrieval depth helps stabilize sub-question answering accuracy and consequently improves overall performance. This reveals a key limitation of our method: incorrect sub-answers may induce a ripple effect throughout the iterative decomposition process—a challenge shared by all multi-hop decomposition-based approaches.

---

Sub-question: Who starred in Duel at Diablo?
Sub-answer: James Garner
Sub-question: Did James Garner also star in Space Cowboys?
Sub-answer: Yes
Sub-question: What year was James Garner born?
sub-answer: 1928
Question: What year was the actor born that starred in both Duel at Diablo and Space Cowboys?
Answer: 1928

---

Table 15: LM responds directly to the original question. When ⟨EOS⟩ is generated in Table 16, the inference chain terminates and the resulting context is used as input to the LM.

---

**Initial prompt:**

Decompose the following question into sub-questions:

What year was the actor born that starred in both Duel at Diablo and Space Cowboys?

- - - - - - - - - - - - - - - - - - - - - - - - - - - - - - - - - - - - - - - - - - - -

Duel at Diablo: Duel at Diablo is a 1966 western film starring James Garner in his first Western since leaving "Maverick" and Sidney Poitier in his first Western.

Sub-question: Who starred in Duel at Diablo?

Sub-answer: James Garner

---

**2-step prompt:**

Decompose the following question into sub-questions:

What year was the actor born that starred in both Duel at Diablo and Space Cowboys?

Sub-question: Who starred in Duel at Diablo?

Sub-answer: James Garner

- - - - - - - - - - - - - - - - - - - - - - - - - - - - - - - - - - - - - - - - - - - -

James Garner: He starred in several television series over more than five decades, including such popular roles as Bret Maverick in the 1950s western comedy series "Maverick" and Jim Rockford in "The Rockford Files"; and played leading roles in more than 50 theatrical films, including "The Great Escape" (1963) with Steve McQueen, Paddy Chayefsky's "The Americanization of Emily" (1964), "Grand Prix" (1966), Blake Edwards' "Victor/Victoria" (1982), "Murphy's Romance" (1985), for which he received an Academy Award nomination, "Space Cowboys" (2000) with Clint Eastwood, and "The Notebook" (2004).

Sub-question: Did James Garner also star in Space Cowboys?

Sub-answer: Yes

---

**3-step prompt:**

Decompose the following question into sub-questions:

What year was the actor born that starred in both Duel at Diablo and Space Cowboys?

Sub-question: Who starred in Duel at Diablo?

Sub-answer: James Garner

Sub-question: Did James Garner also star in Space Cowboys?

Sub-answer: Yes

- - - - - - - - - - - - - - - - - - - - - - - - - - - - - - - - - - - - - - - - - - - -

James Garner: James Garner (born James Scott Bumgarner; April 7, 1928 – July 19, 2014) was an American actor, producer, and voice artist.

Sub-question: What year was James Garner born?

sub-answer: 1928

---

**4-step prompt:**

Decompose the following question into sub-questions:

What year was the actor born that starred in both Duel at Diablo and Space Cowboys?

Sub-question: Who starred in Duel at Diablo?

Sub-answer: James Garner

Sub-question: Did James Garner also star in Space Cowboys?

Sub-answer: Yes

Sub-question: What year was James Garner born?

sub-answer: 1928

- - - - - - - - - - - - - - - - - - - - - - - - - - - - - - - - - - - - - - - - - - - -

⟨EOS⟩

---

Table 16: Case example generated by the sub-question generator. At each step, the input appears above the dotted line, while the black text below shows the output. Red text indicates the answer to the generated sub-question.

**Initial prompt:**
Decompose the following question into sub-questions:
The actress that plays Sarah Hurst in Ëasy Virtueŝtars with Dervla Kirwan in a 2012 supernatural thriller film written by who?

- - - - - - - - - - - - - - - - - - - - - - - - - - - - - - - - - - - - - - - - - - - - -

Charlotte Riley: She is known for her roles as Sarah Hurst in "Easy Virtue" and as Catherine Earnshaw in ITV's adaptation of "Wuthering Heights".
Sub-question: Who plays Sarah Hurst in the 2010 film 'Easy Virtue'?
Sub-answer: Charlotte Riley

**2-step prompt:**
Decompose the following question into sub-questions:
The actress that plays Sarah Hurst in Ëasy Virtueŝtars with Dervla Kirwan in a 2012 supernatural thriller film written by who?
Sub-question: Who plays Sarah Hurst in the 2010 film 'Easy Virtue'?
Sub-answer: Charlotte Riley

- - - - - - - - - - - - - - - - - - - - - - - - - - - - - - - - - - - - - - - - - - - - -

Into the Dark (film): Into the Dark, also known as "I Will Follow You into the Dark", is a 2012 supernatural romance thriller written and directed by Mark Edwin Robinson.
Sub-question: Which 2012 supernatural thriller film stars Dervla Kirwan and Charlotte Riley?
Sub-answer: Into the Dark

**3-step prompt:**
Decompose the following question into sub-questions:
The actress that plays Sarah Hurst in Ëasy Virtueŝtars with Dervla Kirwan in a 2012 supernatural thriller film written by who?
Sub-question: Who plays Sarah Hurst in the 2010 film 'Easy Virtue'?
Sub-answer: Charlotte Riley
Sub-question: Which 2012 supernatural thriller film stars Dervla Kirwan and Charlotte Riley?
Sub-answer: Into the Dark

- - - - - - - - - - - - - - - - - - - - - - - - - - - - - - - - - - - - - - - - - - - - -

Into the Dark (film): Into the Dark, also known as "I Will Follow You into the Dark", is a 2012 supernatural romance thriller written and directed by Mark Edwin Robinson.
Sub-question: Who wrote the 2012 supernatural thriller film 'Into the Dark'?
sub-answer: Mark Edwin Robinson

**4-step prompt:**
Decompose the following question into sub-questions:
The actress that plays Sarah Hurst in Ëasy Virtueŝtars with Dervla Kirwan in a 2012 supernatural thriller film written by who?
Sub-question: Who plays Sarah Hurst in the 2010 film 'Easy Virtue'?
Sub-answer: Charlotte Riley
Sub-question: Which 2012 supernatural thriller film stars Dervla Kirwan and Charlotte Riley?
Sub-answer: Into the Dark
Sub-question: Who wrote the 2012 supernatural thriller film 'Into the Dark'?
Sub-answer: Mark Edwin Robinson

- - - - - - - - - - - - - - - - - - - - - - - - - - - - - - - - - - - - - - - - - - - - -

⟨EOS⟩

Table 17: A failure case induced by sub-question retrieval. An incorrect retrieval result generated from a sub-question triggers a chain reaction, ultimately resulting in an overall failure.

