# OpenReview forum: "MergePRAG: Orthogonal Merging of Passage-experts for Multi-hop Parametric RAG"
_ICLR.cc/2026/Conference — ICLR 2026 Poster_

### Official Review · Reviewer_5C8J · 2025-10-27

**Soundness:** 3
**Presentation:** 3
**Contribution:** 3
**Rating:** 6
**Confidence:** 3

**Summary:**

This paper introduces MergePRAG, a novel framework that extends Parametric Retrieval-Augmented Generation (PRAG) to handle multi-hop reasoning, a limitation of previous methods restricted to single-pass retrieval. The proposed approach sequentially integrates knowledge from multiple retrieved passages into the large language model's parameters during inference. This is achieved through two key innovations: (1) orthogonal continual merging, which uses the Gram-Schmidt process to add new "passage experts" without conflicting with previously integrated knowledge, and (2) critical-layer parameterization, which efficiently injects this new knowledge by updating only a select critical layer, reducing computational cost. Experiments on multi-hop question answering and reasoning-aware knowledge editing tasks demonstrate that MergePRAG consistently outperforms standard RAG, state-of-the-art RAGs, and other parametric adaptation methods.

**Strengths:**

This paper has the following strengths:

- It introduces MergePRAG, the first framework to successfully generalize Parametric Retrieval-Augmented Generation (PRAG) to the more complex multi-hop reasoning setting, addressing a key limitation of prior PRAG methods.
- It proposes a novel "continual merging" technique that uses orthogonal merging to integrate new knowledge without conflicts and critical-layer parameterization to ensure high efficiency, allowing knowledge to accumulate effectively across retrieval steps.
- It performs extensive experiments across multiple benchmarks and LLM backbones, demonstrating that MergePRAG consistently outperforms existing RAG and PRAG baselines in both effectiveness and efficiency.

**Weaknesses:**

This paper has the following weaknesses:

- The evaluation relies solely on token-level metrics (Exact Match and F1), which may not fully capture the semantic quality or factual accuracy of the generated responses. The study would be strengthened by a more holistic, model-based evaluation, such as using an LLM-as-a-judge.

- The experiments are limited to models up to 8B parameters. It remains an open question whether the proposed method's benefits and efficiency gains will scale effectively to current state-of-the-art, much larger models (e.g., 200B+ parameters).

- The analysis identifying optimal "critical layers" for expert injection (Figure 2) is conducted on only one model (LLaMA3.1-8B). It is unclear if this finding generalizes across different model architectures, which is a key assumption for the "critical-layer parameterization" technique.

**Questions:**

- Given the limitations of Exact Match and F1 in capturing semantic quality, can the authors provide justification for omitting a model-based evaluation (like LLM-as-a-judge), or discuss how these metrics adequately validate the model's multi-hop reasoning capabilities?

- The experiments are limited to models under 8B parameters. Can the authors provide any analysis or theoretical argument on whether the efficiency and effectiveness benefits of MergePRAG are expected to scale to current, much larger state-of-the-art models?

- The critical-layer parameterization relies on findings from a single model (LLaMA3.1-8B). Have the authors verified that this "early-to-middle layer" sensitivity generalizes to other architectures? If not, how does this potential model-specificity impact the method's general applicability?

---

> ### Author Response · Authors · 2025-12-01
> **Response for Weakness 1.**
>
> We thank the reviewer for the valuable suggestion regarding the incorporation of model-based evaluation methods such as LLM-as-a-judge. We agree that such evaluations can offer a more comprehensive assessment of semantic quality and factual consistency. However, in the current RAG literature, **Exact Match (EM) and F1 remain the most widely adopted and directly comparable evaluation metrics**, and many state-of-the-art works report results primarily using these measures. To ensure consistency and comparability with prior studies, our main evaluation follows these standard metrics.
>
> Furthermore, because multi-hop QA performance is tightly coupled with token-level alignment to retrieved evidence, EM and F1 are particularly suited for capturing correctness in this setting.
>
> Of course, we also seriously note recent work exploring LLM-as-a-judge for RAG evaluation (e.g., arXiv:2504.15205), which we view as a promising complementary evaluation direction. In future work or an extended version, we plan to incorporate LLM-based evaluation on additional benchmarks. We thank the reviewer again for the constructive suggestion.

---

> ### Author Response · Authors · 2025-12-01
> **Response for Weakness 2.**
>
> We thank the reviewer for raising this important point. We acknowledge that our current experiments are limited to models up to 8B parameters, and that evaluating scalability to much larger models is an open and meaningful direction. Due to practical compute constraints, we were unable to run experiments on 200B+ models, but we believe that MergePRAG is inherently scalable for the following reasons.
>
> - **Model-agnostic design.**
> MergePRAG does not depend on architectural properties specific to small models. Its core components—passage-memory injection and orthogonal continual merging—operate directly on key–value representations, which exist identically across larger transformer architectures.
>
> - **Lightweight hypernetwork.**
> The hypernetwork is extremely small relative to the base model, and its computational and memory overhead grows minimally with model size. This suggests that MergePRAG can naturally extend to larger backbones.
>
> We also recognize the reviewer’s concern regarding whether the hypernetwork must scale appropriately when generating key–value matrices for larger models with higher dimensionality. While the current lightweight hypernetwork serves as a minimal yet fully functional instantiation designed for efficiency, it can be scaled up (e.g., by adding layers or increasing hidden dimensionality) to match the requirements of larger transformer architectures. Considering this point, although 200B+ evaluations are infeasible under our current compute budget, we plan to extend our experiments to larger models such as LLaMA-30B, which would provide useful supporting evidence in an extended version or future work.
>
> In summary, while our compute budget prevents direct evaluation on 200B+ models, the design of MergePRAG suggests strong potential for scaling to larger architectures. We plan to explore this direction as resources permit, including extensions to larger models such as LLaMA-30B.

---

> ### Author Response · Authors · 2025-12-01
> **Response for Weakness 3.**
>
> We thank the reviewer for raising this important point. We agree that, given differences in model architectures and training paradigms, layer-scanning analysis should be conducted across multiple models to validate the assumption underlying critical-layer parameterization.
>
> In the revised version of the paper, we have added layer-scanning experiments for both LLaMA3.1-8B and Qwen2.5-7B across several datasets (**Figures 2–7**). The results show clear model-dependent patterns: for LLaMA3.1-8B, early-layer injection yields the weakest performance, whereas for Qwen2.5-7B, the least effective injection occurs at the final layers. These observations support the necessity of performing model-specific layer-scanning.
>
> We note that prior work, such as ROME[1] and MEMIT[2], similarly conducts causal-tracing and layer-level analyses across different architectures, further motivating our approach. In this sense, our method can be viewed as a ***locate-then-inject*** framework, drawing inspiration from these knowledge-editing methodologies.
>
> We are also interested in exploring more advanced causal mediation and mechanistic interpretability techniques to better understand where passage-level experts should be injected. We appreciate the reviewer for highlighting this point.
>
>
>
> ### **Reference**:
> [1] Meng K, Bau D, Andonian A, et al. Locating and editing factual associations in GPT. *Advances in Neural Information Processing Systems*, 2022, 35: 17359-17372.
> [2] Meng K, Sharma A S, Andonian A J, et al. Mass-Editing Memory in a Transformer. In *ICLR*, 2023.

---

> ### Author Response · Authors · 2025-12-01
> **Response for Question 1.**
>
> We thank the reviewer for raising this question. Across the multi-hop QA datasets used in our experiments (HotpotQA and 2WikiMhQA), the questions fall into several well-defined categories:
>
> 1. **Multi-hop reasoning questions** — requiring aggregation of information from multiple paragraphs.
> Examples include:
>     - “Which person associated with \_\_\_ was born earlier?”
>     - “What is the nationality of the author of \_\_\_?”
>
>
> 2. **Bridge-type questions** — requiring reasoning through an intermediate entity (A → B → answer).
> For example:
>     - “In which country was the author of \emph{Harry Potter} born?”
> which requires first identifying “J. K. Rowling” and then determining her birth country.
>
> 3. **Comparison questions** — requiring comparison of attributes across entities.
> Example:
>     - “Who was born earlier, A or B?”
>
> For these question types, the correct answers are typically entities or short spans that appear directly in the retrieved context. Therefore, token-level metrics such as Exact Match (EM) and F1 are well aligned with evaluating multi-hop reasoning in this setting: they directly measure whether the model correctly identifies and generates the grounded answer span.
>
> Moreover, **EM and F1 remain the standard and most widely adopted evaluation metrics** in the RAG and multi-hop QA literature, enabling reliable and fair comparison with prior work. Thus, we follow these established metrics for consistency and comparability.
>
> Of course, as replied previously to the weakness part, we acknowledge that model-based evaluation methods (e.g., LLM-as-a-judge) can offer additional semantic insight. Recent work, such as work [1],  explores this direction, and we agree that such evaluation is valuable.
>
> Due to time constraints, we were unable to include LLM-as-a-judge evaluation within the rebuttal period, but we plan to incorporate such analysis in an extended version or in future work. Thank you again for your valuable comment.
>
> ### **Reference**
> [1] Thakur N, Pradeep R, Upadhyay S, et al. Support evaluation for the TREC 2024 RAG track: Comparing human versus LLM judges[J]. arXiv preprint arXiv:2504.15205, 2025.

---

> ### Author Response · Authors · 2025-12-01
> **Response for Question 2.**
>
> We thank the reviewer for this important question. Below, we provide both theoretical and practical arguments for why the efficiency and effectiveness benefits of MergePRAG are expected to scale to much larger Transformer models, together with caveats and deployment heuristics.
>
> 1. **Transformer universality — injected K/Vs integrate naturally.**
> Modern SOTA LLMs, regardless of size, remain Transformer-based and compute attention through the same key/value (K,V) and query (Q) projections at each layer. Injecting paragraph representations as additional K/V vectors directly interfaces with this universal attention mechanism. Since attention is the core module responsible for cross-context integration, providing external K/Vs is an architecture-agnostic method for biasing the model’s reasoning, and this property is independent of model scale.
>
> 2. **Hypernetwork-based injection is size-friendly.**
> The hypernetwork produces a compact set of paragraph vectors, and its parameter count is orders of magnitude smaller than that of the base model. Thus, the compute required to generate and inject K/Vs is negligible relative to a single forward pass of a 200B+ model. As model size increases, the dominant cost remains the base model forward pass, so the relative overhead of MergePRAG shrinks further.
>
> 3. **Critical-layer parameterization remains meaningful at scale.**
> Across Transformer architectures of different sizes, layers consistently exhibit functional specialization (e.g., lexical processing, semantic integration, task-specific projection). The existence of a “critical layer’’ where external evidence integrates most effectively is therefore architectural rather than model-size–specific. Our lightweight scanning procedure provides a practical way to identify such layers, and its cost is minimal: it requires only forward-pass evaluation (e.g., perplexity), not retraining. As models grow larger, this probing becomes even cheaper relative to full fine-tuning.
>
> 4. **Efficiency argument.**
> For a base Transformer, the dominant forward cost scales as $\mathcal{O}(L \, n \, d^{2})$. Injecting $k$ additional K/V vectors modifies attention computation at only the chosen layer(s) and has cost $\mathcal{O}(L \, k \, d)$, which is small when $k \ll d$.
> Thus, the relative extra compute introduced by MergePRAG becomes \emph{smaller} as model width $d$ and model size increase.
>
> On theoretical and computational grounds, the core mechanisms of MergePRAG—K/V injection, hypernetwork-based generation, and orthogonal continual merging—are architecture-agnostic primitives that extend naturally to larger Transformers, with minimal and diminishing overhead. We therefore expect MergePRAG’s effectiveness and efficiency benefits to generalize to 200B+ models.
>
> As noted previously, we plan to explore this direction with larger backbones such as LLaMA-30B to empirically validate hypernetwork and merging scalability when resources permit. We thank the reviewer again for the valuable comment.

---

> ### Author Response · Authors · 2025-12-01
> **Response for Question 3.**
>
> Thank you for raising this important point. In **Appendix A: Layer Scanning Experiments for Critical-Layer Parameterization**, we have added layer-scanning analyses for both LLaMA3.1-8B and Qwen2.5-7B across multiple datasets (see **Figures 2–7**).
>
> These results confirm that sensitivity patterns indeed differ across architectures: early-to-middle layers are most effective for LLaMA3.1-8B, whereas Qwen2.5-7B exhibits its weakest performance at the final layers. This reinforces the necessity of conducting model-specific layer scanning.
>
> Regarding general applicability, we recommend performing a lightweight layer-scanning procedure for any new backbone. Because the hypernetwork is small and the scanning requires only forward-pass evaluation, the procedure is computationally efficient and introduces negligible overhead. Thus, the critical-layer parameterization technique remains practical and broadly applicable across different model architectures.

---

### Official Review · Reviewer_YcxX · 2025-10-31

**Soundness:** 3
**Presentation:** 3
**Contribution:** 3
**Rating:** 6
**Confidence:** 3

**Summary:**

This paper introduces MergePRAG, a novel framework that extends Parametric RAG (PRAG) to multi-hop question answering. The key innovation is a continual merging mechanism that sequentially integrates retrieved passages into LLM parameters through orthogonal merging (using Gram-Schmidt process) and critical-layer parameterization. The method shows consistent improvements over existing RAG and PRAG baselines on multiple multi-hop QA datasets.

**Strengths:**

1. This paper investigates a critical research question: how to inject retrieved knowledge into parameters for multi-hop QA? The motivation is novel, and the proposed methods address the concerns of conflicts during merging and maintaining lightweight.
2. The experiments show advantages on multi-hop QA datasets, comparing with a wide range of methods.
3. Some augmentation designs like only modifying the critical layer, and shared basis low-rank parameters make sense.

**Weaknesses:**

1. The advantages need further clarification. When compared to other PRAG methods like DyPRAG, how does the pipeline work? Is the sub-question generator used consistently across all PRAG baselines? What are the exact architectural and procedural differences that lead to performance gains?

2. The significance of the method lies in the passage merging that avoids knowledge conflict, compared to other PRAG methods. First, the severity of the knowledge conflict may need illustration. The paper doesn't quantify how often or how severely knowledge conflicts occur when passages from different sub-questions are merged. Second, if the gram-schmidt orthonormalization is ablated, will the performance significantly decrease? There is no empirical evidence showing the actual conflict patterns in the retrieved passages

3. There is no analysis on the significance of critical layer selection, and shared basis low-rank parameters. Missing comparison of different merging strategies. Computational overhead comparison with baselines would strengthen the efficiency claims.

4. The paper would benefit from error analysis showing failure cases.

**Questions:**

1. Why does lower perplexity indicate higher sensitivity to expert injection?

---

> ### Author Response · Authors · 2025-12-01
> **Response for weakness 1.**
>
> We thank the reviewer for the insightful question. Below we clarify the architectural and procedural differences between MergePRAG and prior PRAG methods such as DyPRAG, as well as the sources of our performance gains.
>
> 1. **Pipeline differences between DyPRAG and MergePRAG.**
> As summarized in **Section~3.2**, DyPRAG requires a two-stage training pipeline:
> (i) a ***Document Augmentation*** step that generates synthetic question–answer pairs for each document; and
> (ii) a ***distillation*** stage in which the hypernetwork learns to encode document knowledge by answering these QA pairs and producing LoRA modules.
>
> Thus, DyPRAG relies on document-level QA generation and LoRA-based parameter patches, and its effectiveness is strongly tied to the quality of the generated QA pairs.
>
> In contrast, MergePRAG follows a fundamentally different and more streamlined design. MergePRAG maps each passage directly into a passage-specific \emph{memory} representation using a lightweight hypernetwork that operates only on the in-context passage itself—without requiring document-level QA generation, distillation, or LoRA patch construction. These memory vectors are
> injected into the model’s critical layer and subsequently merged through orthogonal continual merging. This design completely removes DyPRAG’s multi-step augmentation–distillation pipeline and substantially reduces computational overhead.
>
> 2. **Sub-question generator consistency across baselines.**
> We confirm that all PRAG-style baselines requiring sub-question generation use the same sub-question generator in our experiments. This ensures strict comparability and isolates the effect of our proposed parameterization and merging mechanisms.
>
> 3. **Sources of MergePRAG’s performance gains.**
> The improvements observed in **Table 3** primarily arise from:
> (i) the direct injection of compact passage-specific memory vectors, which provide a focused representation of the passage without requiring synthetic QA-based supervision; and
> (ii) the orthogonal continual merging mechanism, which preserves complementary information across hops while mitigating interference among parameter patches.  Compared with LoRA-based patches produced by DyPRAG, these memory vectors are
> more efficient, more stable across hops, and better aligned with the fine-grained information present in retrieved passages.
>
> **Table 3: Comparison of MergePRAG+ and MultihopRAG with fine-tuning (without hypernetwork) under LLaMA3.1-8B**
> | Training type                                        | HotpotQA EM | HotpotQA F1 | 2WikiMhQA EM | 2WikiMhQA F1 |
> |------------------------------------------------------|-------------|-------------|--------------|--------------|
> | MultihopRAG$_{\|SP\| = 1}$  (w/o finetuning)              | 37.80       | 47.56       | 23.30        | 35.59        |
> | MultihopRAG$_{\|SP\|=1}$ (finetuning: [sq → sa])       | 43.70       | 50.15       | 58.10        | 62.57        |
> | MultihopRAG$_{\|SP\|=1}$ (finetuning: [(P_golden, sq) → sa]) | 40.10 | 46.79       | 60.30        | 62.04        |
> | MergePRAG + $_{\|SP\|=1}$                              | 47.40       | 55.29       | 65.60        | 70.54        |
>
> 4. **Conceptual distinction.**
> MergePRAG is not merely an extension of existing PRAG methods to the multi-hop setting. It introduces a lightweight hypernetwork that produces key–value memory parameters directly from retrieved passages and explicitly separates parameterization from merging. This disentanglement allows the hypernetwork to focus purely on parameterization, while the merging mechanism handles multi-hop accumulation in a principled way.
>
> We will continue to refine the manuscript and expand the comparison details to ensure clarity. We thank the reviewer again for this helpful comment.

---

> ### Author Response · Authors · 2025-12-01
> **Response for Weakness 2.**
>
> Thank you for highlighting the importance of validating knowledge conflict and the role of Gram–Schmidt orthonormalization in our merging strategy. We agree that knowledge conflict among retrieved passages is an important factor, and we plan to include a more explicit quantitative analysis of conflict severity and frequency in an extended version of the paper.
>
> In response to the reviewer’s question about the necessity of orthonormalization, we added a new section, **4.3.3 Effect of the
> Merging Methods**, which includes an ablation study directly comparing Gram–Schmidt orthonormalization with several alternative merging strategies.
>
> As shown in **Table~6**, when merging passage memories from different sub-questions, orthogonal merging consistently outperforms TIES merging and other non-orthogonal strategies such as additive, arithmetic-mean, and concat merging. The fact that these non-orthogonal baselines underperform the orthonormalization-based method suggests that knowledge conflicts do arise
> when combining multiple passage memories -- even in the combined setting where in-context passages are also provided.
>
> **Table 6: MergePRAG+ / HotpotQA — Performance comparison between different merging methods
> (Setting: $\|SP\|$=1)**
>
> | $\mathsf{Merge}_{seq}$       | EM    | F1    |
> |------------------------------|-------|-------|
> | $\blacktriangle$             | 36.20 | 43.47 |
> | $\bullet$                    | 48.20 | 55.04 |
> | $\blacklozenge$              | 46.40 | 54.07 |
> | $\blacktriangledown$         | 48.20 | 54.95 |
> | $\blacksquare$               | 48.80 | 55.53 |
>
> **Legend:**
> - $\blacktriangle$: Adding merging
> - $\bullet$: Arithmetic mean merging
> - $\blacklozenge$: TIES merging
> - $\blacktriangledown$: Connection merging
> - $\blacksquare$: Gram–Schmidt orthogonalization merging
>
>
> Furthermore, **Table~5** shows that orthogonal merging achieves the most robust improvements across a range of settings, often yielding more than 1\% EM gain over arithmetic pooling even when in-context passages are used. This provides additional empirical evidence that restructuring passage vectors into orthogonal subspaces helps mitigate interference and retain complementary information across hops.
>
> **Table 5: Performance comparison between different merging methods for $\mathsf{Merge}_{inner}$ and $\mathsf{Merge}\_{seq}$ varying \|SP\|**
>
> | $\mathsf{Merge}_{inner}$ | $\mathsf{Merge}_{seq}$ | \|SP\|=2 |     | \|SP\|=4 |     | \|SP\|=6 |     | \|SP\|=8 |     | \|SP\|=10 |     | \|SP\|=12 |     |
> |--------------------------|------------------------|---------|-----|---------|-----|---------|-----|---------|-----|----------|-----|----------|-----|
> |                          |                        | EM      | F1  | EM      | F1  | EM      | F1  | EM      | F1  | EM       | F1  | EM       | F1  |
> | $\bullet$                | $\bullet$               | 50.20 | 58.07 | 51.40 | 60.35 | 51.40 | 59.82 | 52.00 | 60.86 | 55.00 | 62.84 | 54.40 | 62.76 |
> | $\bullet$                | $\blacksquare$          | 50.60 | 58.06 | 52.00 | 60.64 | 52.00 | 60.21 | 53.00 | 61.32 | 55.40 | 63.40 | 54.60 | 62.64 |
> | $\blacksquare$           | $\bullet$               | 50.40 | 58.14 | 51.60 | 60.13 | 51.40 | 59.63 | 52.60 | 61.46 | 54.80 | 62.80 | 54.60 | 62.76|
> | $\blacksquare$           | $\blacksquare$          | **50.80** | **58.23** | **52.40** | **60.67** | **52.40** | **60.67** | **53.40** | **61.77** | **55.60** | **63.45** | **55.00** | **62.93** |
>
> **Legend:**
> - $\bullet$：Arithmetic mean merging
> - $\blacksquare$：Gram–Schmidt orthogonalization merging
>
>
> We acknowledge that further analysis -- especially under the purely parametric setting of MergePRAG (without in-context passages) -- would offer deeper insight into the nature of knowledge conflicts. We plan to extend these experiments and include conflict statistics and visualization in an extended version.
>
> In summary, the ablation results in **Tables~5 and ~6** clearly demonstrate that Gram–Schmidt orthonormalization contributes meaningfully to the gains of MergePRAG+. We appreciate the reviewer’s feedback and will prioritize expanding the quantitative analysis of knowledge conflict in an extended version of the work.

---

> ### Author Response · Authors · 2025-12-01
> **Response for Weakness 3.**
>
> Thank you for raising the need for deeper analysis of critical-layer selection, shared basis low-rank parameters, merging strategies, and computational overhead. We have addressed each of these concerns in the revised manuscript.
>
> First, in **Appendix A: Layer Scanning Experiments for Critical-Layer Parameterization**, we provide a detailed analysis of the critical layer selection process across different models and datasets. These experiments demonstrate clear, model-dependent sensitivity patterns and motivate the necessity of identifying the injection layer empirically.
>
> Second, regarding parameterization, we modified the hypernetwork in the revision to generate key–value memory representations. We include a new ablation study in **Section 4.3.5: Effect of the Number of Key–Value Vectors** (Table~8), which examines how varying the number of shared key–value vectors affects expressive capacity and downstream performance. The results show that
> the number of vectors plays a significant role in determining the richness of the injected paragraph representation.
>
> **Table 8: Ablation on the Number of Passage Vectors $num_{kv}$ for LLaMA3.1-8B**
>
> | $k$ (i.e., $num_{kv}$) | | HotpotQA  | |  |  |2WikiMhQA  |  |  |
> |-------------------------|--------|--------|--------|--------|---------|---------|---------|---------|
> |                           | \|SP\|=1 |        | \|SP\|=4 |         | \|SP\|=1  |          | \|SP\|=4  |           |
> |                           | EM       | F1       | EM       | F1       | EM        | F1        |   EM        | F1        |
> | 1                         | 45.60    | 52.67    | 49.00    | 58.24    | 62.40     | 67.89     | 69.00     | 75.21     |
> | 2                         | 45.60    | 52.67    | 51.20    | 59.20    | 63.80     | 69.01     | 69.00     | 76.32     |
> | 4                         | 45.60    | 52.86    | 50.80    | 58.88    | 64.00     | 69.37     | 71.20     | 77.09     |
> | 8                         | 46.40    | 54.25    | 49.40    | 58.39    | 65.90     | 70.93     | 72.00     | 78.09     |
> | 16                        | 48.80    | 55.53    | 52.40    | 60.67    | 66.30     | 71.05     | 73.20     | 79.34     |
>
>
> Third, in **Section 4.3.3: Effect of the Merging Methods**, we compare several merging strategies. (**As shown in table~5 and ~6 of [Response for W 2.](https://openreview.net/forum?id=FSL1J2gmJV&noteId=3XSfcUWAux)**)  The Gram–Schmidt orthonormalization approach consistently outperforms alternatives such as additive, arithmetic-mean, and concat merging, providing empirical evidence of its importance in mitigating knowledge conflicts across sub-questions.
>
> Finally, we added **Section E.1: Efficiency Analysis** in the appendix (Table~11), where we quantify the computational overhead of the proposed hypernetwork-based passage parameterization. The analysis shows that the overhead is minimal relative
> to the remaining components of the system.
>
> **Table 11: Efficiency analysis of MergePRAG using the LLaMA3.1-8B model on the HotpotQA dataset**
> | Method | Generate Passage Memory | Response $sa$ | Average Time |
> |:------:|:---------------------:|:------------:|:------------:|
> | $\text{RAG}_{\|SP\|=1}$ | - | - | 0.712s |
> | $\text{RAG-CoT}_{\|SP\|=1}$ | - | - | 6.389s |
> | $\text{MergePRAG+}_{\|SP\|=1}$ | 0.001s | 0.259s | 2.517s |
>
>
> We appreciate the reviewer’s helpful suggestions and believe these additions significantly strengthen the clarity and completeness of the manuscript.

---

> ### Author Response · Authors · 2025-12-01
> **Response for Weakness 4.**
>
> Thank you for the helpful suggestion. We have added an error analysis in **Appendix H: Case Study** (Tables 16–17). One representative failure case occurs when an early sub-question retrieves an incorrect passage. This leads to an incorrect sub-answer, which then propagates through subsequent sub-questions and results in a cascading failure in the final answer.
>
> This observation is consistent with our $|SP| > 1$ experiments (Tables 5 and 7), where increasing the number of relevant passages per sub-question consistently improves performance. These results highlight that accurate retrieval and correct sub-question answering are crucial for multi-hop reasoning, and early errors can have amplified downstream effects.
>
> **Table 5: Performance comparison between different merging methods for $\mathsf{Merge}_{inner}$ and $\mathsf{Merge}\_{seq}$ varying \|SP\|**
> | $\mathsf{Merge}_{inner}$ | $\mathsf{Merge}_{seq}$ | \|SP\|=2 |     | \|SP\|=4 |     | \|SP\|=6 |     | \|SP\|=8 |     | \|SP\|=10 |     | \|SP\|=12 |     |
> |--------------------------|------------------------|---------|-----|---------|-----|---------|-----|---------|-----|----------|-----|----------|-----|
> |                          |                        | EM      | F1  | EM      | F1  | EM      | F1  | EM      | F1  | EM       | F1  | EM       | F1  |
> | $\bullet$                | $\bullet$               | 50.20 | 58.07 | 51.40 | 60.35 | 51.40 | 59.82 | 52.00 | 60.86 | 55.00 | 62.84 | 54.40 | 62.76 |
> | $\bullet$                | $\blacksquare$          | 50.60 | 58.06 | 52.00 | 60.64 | 52.00 | 60.21 | 53.00 | 61.32 | 55.40 | 63.40 | 54.60 | 62.64 |
> | $\blacksquare$           | $\bullet$               | 50.40 | 58.14 | 51.60 | 60.13 | 51.40 | 59.63 | 52.60 | 61.46 | 54.80 | 62.80 | 54.60 | 62.76|
> | $\blacksquare$           | $\blacksquare$          | **50.80** | **58.23** | **52.40** | **60.67** | **52.40** | **60.67** | **53.40** | **61.77** | **55.60** | **63.45** | **55.00** | **62.93** |
>
> **Legend:**
> - $\bullet$：Arithmetic mean merging
> - $\blacksquare$：Gram–Schmidt orthogonalization merging
>
> **Table 7: Performance of MergePRAG+ on HotpotQA and 2WikiMHQA with varying numbers of retrieved passages ($|SP|$) per sub-question. Increasing $|SP|$ provides broader evidence for answering each sub-question, which can improve overall QA accuracy.**
>
> | $\|SP\|$ | HotpotQA EM | HotpotQA F1 | 2WikiMhQA EM | 2WikiMhQA F1 |
> |:--------:|:-----------:|:-----------:|:------------:|:------------:|
> | 1        | 48.80       | 55.53       | 66.30        | 71.05        |
> | 2        | 50.80       | 58.23       | 71.40        | 76.94        |
> | 3        | 52.00       | 59.50       | 73.10        | 79.06        |
> | 4        | 52.40       | 60.67       | 73.20        | 79.34        |
>
>
> We appreciate the reviewer’s comment and believe that the added error analysis provides useful insight into the behavior and limitations of our method.

---

> ### Author Response · Authors · 2025-12-01
> **Response for Question 1.**
>
> Thank you for the question regarding the relationship between perplexity and layer sensitivity. In our layer-scanning experiments, we inject the paragraph-specific key–value vectors $(K, V)$ at each layer and evaluate the model’s response by computing perplexity on the target answer. Perplexity is defined as:
>
> $$
> \mathrm{PPL} = \exp\left(-\frac{1}{N} \sum_{i=1}^{N} \log p(x_i)\right),
> $$
>
> where $p(x_i)$ is the model’s predicted probability for the correct token $x_i$. A lower perplexity therefore corresponds to a higher likelihood assigned to the correct answer tokens.
>
> When expert information is injected at a particular layer, a larger reduction in perplexity indicates that the injected representation has a stronger influence on the model’s internal computation, making the model more confident in producing the correct answer.
> Conversely, if perplexity changes only minimally, the injection has little effect, suggesting that the layer is less sensitive to
> external expert information.
>
> Thus, lower perplexity serves as an empirical indicator of higher sensitivity to vector injection: it reflects that the paragraph vectors are being integrated more effectively at that layer. This forms the basis of our critical-layer selection procedure. We note that exploring additional metrics and interpretation tools to analyze layer importance is an interesting direction for future work.

---

### Official Review · Reviewer_EL8F · 2025-10-31

**Soundness:** 2
**Presentation:** 3
**Contribution:** 3
**Rating:** 2
**Confidence:** 4

**Summary:**

This paper proposes MergePRAG, a novel extension of Parametric RAG (PRAG) that enables multi-hop reasoning by orthogonally merging passage-specific experts into a language model’s parameter space. The method introduces an orthogonal merging mechanism, critical-layer parameterization, and a hypernetwork-based low-rank adaptation to efficiently absorb multi-hop retrieved knowledge. The idea is timely and original, addressing an important limitation of current PRAG systems that struggle with multi-hop reasoning. The paper is generally well-written and conceptually interesting, with promising empirical results.

**Strengths:**

The idea of orthogonal merging of passage experts is innovative and aligns well with current trends in retrieval-augmented and parametric reasoning research.

The approach is conceptually neat, allowing models to accumulate knowledge from multiple retrieval steps while minimizing interference.

The authors evaluate the method on multiple multi-hop QA and knowledge-editing datasets, showing that MergePRAG achieves competitive or even superior performance compared to prior RAG and PRAG variants.

**Weaknesses:**

1. Inconsistency and Missing Entries in Table 1
Table 1 looks quite confusing. Several entries (e.g., DyPRAG’s EM scores) are missing. For Qwen2.5-7B, there are no PRAG or DyPRAG baselines, which makes it unclear whether the claimed improvement of MergePRAG+ is significant or merely due to the absence of strong comparisons. Moreover, Table 1 reports MergePRAG+ results only—there seems to be no result for the base MergePRAG. It is also unclear whether MultiHopRAG corresponds to MergePRAG or another baseline. Overall, the table lacks a fair and consistent setting for comparison, casting doubt on the validity of the experimental conclusion

2. Effectiveness of Knowledge Injection (Table 4) According to Table 4, the performance of MergePRAG appears to drop substantially under certain configurations. This raises concerns about the effectiveness and stability of the proposed parametric knowledge injection. The authors should discuss why this degradation happens and whether it reflects limitations in the merging mechanism or the hypernetwork parameterization.

3. The paper does not include a natural baseline where all retrieved documents are merged into a single LoRA adaptation (without orthogonal decomposition). Such a comparison would be essential to demonstrate that orthogonal merging is the key factor, rather than simply adding more LoRA parameters. Without this, it is hard to isolate the contribution of the proposed mechanism.

4. The choice of “critical layers” is somewhat arbitrary and lacks theoretical or empirical justification. The method appears to require layer-by-layer scanning to find the best injection point, which seems neither elegant nor scalable. Moreover, it is unclear whether different documents or tasks share the same optimal layer, or whether the critical layer should adapt dynamically. This design choice needs clearer motivation and analysis.

5. The method seems tuned for specific architectures (LLaMA-3.1-8B and Qwen-2.5-7B). There is little evidence that the approach generalizes across other model types, sizes, or training regimes. Since the critical layer and merging behavior may differ per model, the generalization ability of MergePRAG remains uncertain.

**Questions:**

none

---

> ### Author Response · Authors · 2025-12-01
> **Response for Weakness 1. Part (1/2)**
>
> We thank the reviewer for pointing out the inconsistencies and missing entries in Table~1. We provide detailed clarification below.
>
> 1. **Missing EM scores for DyPRAG.**
> DyPRAG relies on document augmentation, where documents are converted into model-generated QA pairs. The generated answers follow a model-specific style, making Exact Match (EM) an unreliable evaluation metric for DyPRAG. The original
> DyPRAG paper therefore reports F1 only, and we follow the same practice to ensure fairness and consistency.
>
> 2. **Missing DyPRAG baselines for Qwen2.5-7B.**
> The DyPRAG paper does not provide QA-pair generation results for MuSiQue, nor does it release training configurations or checkpoints for Qwen2.5-7B. As a result, reproducing DyPRAG on these settings is not feasible in a fair manner. Nevertheless,
> based on DyPRAG’s reported F1 scores under LLaMA3.1-8B, MergePRAG+ consistently outperforms DyPRAG across HotpotQA and 2WikiMhQA, suggesting that DyPRAG’s performance is strongly constrained by the quality of its generated QA pairs.
> We understand the reviewer’s concern regarding comparisons with PRAG and DyPRAG. We would like to clarify that we made every effort to include these baselines whenever reproducible configurations were available, and we had no intention of relying on weaker baselines. Our goal was to evaluate against the strongest applicable baselines so that the advantages of the proposed method could be demonstrated clearly and fairly. Importantly, MergePRAG+ achieves consistent improvements over PRAG and DyPRAG in their original, reproducible settings, and these gains are not diminished by the absence of certain (not very major) missing entries.
> In addition, we provide our own baseline, MultiHopRAG, which performs multi-hop reasoning without any parametric injection. The key difference between MultiHopRAG and MergePRAG+ is the use of passage-specific parameterization. The improvements observed when comparing MergePRAG+ to MultiHopRAG therefore directly reflect the benefit of injecting parameterized knowledge rather than relying solely on in-context information.
>
> 3. **Missing MergePRAG (base) results in Table 1.**
> We report both MergePRAG and MergePRAG+ in Table\~4. MergePRAG injects only paragraph vectors and does not use retrieved passages during inference; therefore, it is not directly comparable with advanced in-context RAG baselines shown in Table\~1. Nonetheless, Table\~4 shows that MergePRAG alone already outperforms vanilla RAG, indicating that the passage-parameterization mechanism is effective even without in-context evidence.
> While MergePRAG alone is not guaranteed to outperform all strong RAG variants, we highlight that the **combined setting MergePRAG+** achieves strong gains over state-of-the-art RAG systems. These improvements are driven specifically by the
> proposed parameterization mechanism, which provides benefits that simpler parameterization approaches do not achieve.
>
> **Table 4: Comparison of MergePRAG and MergePRAG+ under LLaMA3.1-8B.
> $|SP|=0$ denotes MergePRAG, which does not use in-context passages as prompts during inference.**
>
> | Inference type | HotpotQA EM | HotpotQA F1 | 2WikiMhQA EM | 2WikiMhQA F1 |
> |:--------------:|:-----------:|:-----------:|:------------:|:------------:|
> | $\text{RAG}_{\|SP\|=1}$ | 21.60 | 36.67 | 4.90 | 17.36 |
> | $\text{MergePRAG}_{\|SP\|=0}$ | 28.40 | 35.52 | 45.60 | 50.06 |
> | $\text{MergePRAG +}_{\|SP\|=1}$ | 48.80 | 55.53 | 66.30 | 71.05 |

---

> ### Author Response · Authors · 2025-12-01
> **Response for Weakness 1. Part (2/2)**
>
> 4. **Clarification regarding MultiHopRAG.**
> MultiHopRAG is **not** equivalent to MergePRAG. As described in **Section~3.1**, MultiHopRAG is a baseline that performs iterative multi-hop retrieval and generation without any parametric injection. In Table\~3, we compare multiple MultiHopRAG training configurations against MergePRAG+. The improvements over MultiHopRAG show that the gains of MergePRAG+ come from the injected passage-specific parameters rather than changes in the multi-hop pipeline.
>
> **Table 3: Comparison of MergePRAG+ and MultihopRAG with fine-tuning (without hypernetwork) under LLaMA3.1-8B**
> | Training type                                        | HotpotQA EM | HotpotQA F1 | 2WikiMhQA EM | 2WikiMhQA F1 |
> |------------------------------------------------------|-------------|-------------|--------------|--------------|
> | MultihopRAG$_{\|SP\| = 1}$  (w/o finetuning)              | 37.80       | 47.56       | 23.30        | 35.59        |
> | MultihopRAG$_{\|SP\|=1}$ (finetuning: [sq → sa])       | 43.70       | 50.15       | 58.10        | 62.57        |
> | MultihopRAG$_{\|SP\|=1}$ (finetuning: [(P_golden, sq) → sa]) | 40.10 | 46.79       | 60.30        | 62.04        |
> | MergePRAG + $_{\|SP\|=1}$                              | 47.40       | 55.29       | 65.60        | 70.54        |
>
> Overall, while we understand the reviewer’s concern, we believe the revised paper now provides a clearer and more consistent comparison setting. We have updated the manuscript to explicitly clarify which baselines are applicable under each model–dataset configuration, thereby strengthening the validity of our experimental conclusions.
>
> Empirically, we would also like to emphasize the significance of the improvements achieved by the combined setting MergePRAG+. MergePRAG serves as the core parameterization mechanism that enables these gains, and the performance improvements observed in MergePRAG+ stem specifically from the strength of this parameterization. These benefits are not easily attainable
> with simpler or more naive parameterization approaches. We highlight this aspect in the revised paper to ensure that the empirical value and contribution of MergePRAG+ are not overlooked.
>
> Of course, we will continue to refine and clarify our experimental settings, and we plan to further extend our experimental results in an extended version.
>
> Thank you again for this valuable comment.

---

> ### Author Response · Authors · 2025-12-01
> **Response for Weakness 2.**
>
> Thank you for raising this important point. We apologize for any confusion  regarding the interpretation of Table\~4. We would like to clarify that the  results reported for MergePRAG in Table\~4 correspond to a setting in which  ***no external passages are provided as in-context prompts at inference time***. In this configuration, MergePRAG must answer questions solely using the passage-specific parametric memory generated by the hypernetwork. This context-free evaluation is intentional and is designed to isolate and measure the representational capacity of the injected knowledge.
>
> **Table 4: Comparison of MergePRAG and MergePRAG+ under LLaMA3.1-8B.
> $|SP|=0$ denotes MergePRAG, which does not use in-context passages as prompts during inference.**
>
> | Inference type | HotpotQA EM | HotpotQA F1 | 2WikiMhQA EM | 2WikiMhQA F1 |
> |:--------------:|:-----------:|:-----------:|:------------:|:------------:|
> | $\text{RAG}_{\|SP\|=1}$ | 21.60 | 36.67 | 4.90 | 17.36 |
> | $\text{MergePRAG}_{\|SP\|=0}$ | 28.40 | 35.52 | 45.60 | 50.06 |
> | $\text{MergePRAG +}_{\|SP\|=1}$ | 48.80 | 55.53 | 66.30 | 71.05 |
>
> To provide a meaningful reference point, Table\~4 also includes $RAG_{\|DP\|=1}$, a naïve RAG configuration that uses a simple prompt with one retrieved passage. Notably, MergePRAG outperforms $RAG_{\|DP\|=1}$, despite the latter having access to an actual in-context passage during inference. This indicates that the injected passage memory carries substantive information and can support question answering even without retrieved context, demonstrating the effectiveness of the proposed parametric knowledge injection.
>
> The performance gap between MergePRAG and MergePRAG+ should therefore ``not’’ be interpreted as a limitation of the merging mechanism or the hypernetwork. Rather, it reflects the fact that MergePRAG operates without any in-context evidence, whereas MergePRAG+ restores the full RAG pipeline by combining retrieved passages with injected paragraph vectors—naturally leading to
> stronger performance.
>
> In summary, the lower scores of MergePRAG arise from a deliberately context-``free’’ evaluation setup, not from instability or shortcomings in the merging strategy or hypernetwork parameterization.
>
> At this stage, we would like to clarify that we do not intend to claim that the purely parametric setting (MergePRAG without in-context passages) can replace standard in-context RAG or other advanced RAG methods. Rather, our position is that passage parameterization is a complementary mechanism that can provide additional gains on top of in-context RAG. Accordingly,
> **MergePRAG+ is the final proposed run**, and MergePRAG serves as the underlying parameterization component that enables MergePRAG+ to achieve the notable improvements observed in Table\~1—both over strong in-context RAG baselines and over existing parameterization methods such as PRAG and DyPRAG.
>
> The comparison between MergePRAG and MergePRAG+ in Table\~4 is intended to isolate the effect of adding in-context passages. Therefore, the difference between the two should not be viewed as a “performance drop’’ of MergePRAG, but rather as a ***performance gain*** obtained when in-context passages are added directly to the prompt. This gain is naturally large: in-context passages provide a complete, verbatim sequence of knowledge, allowing the model to access detailed information without risk of forgetting. In contrast, parameterized knowledge—although compact and efficient—may lose some fine-grained details when compressed into a memory representation. This is precisely why combining both forms of knowledge in MergePRAG+ yields stronger performance.
>
> Despite operating without any in-context passages, MergePRAG alone still outperforms a naive RAG configuration ($RAG_{\|DP\|=1}$) that uses an actual retrieved passage at inference time (Table\~4 of the revised version). This is a notable and interesting result that demonstrates the strength of the proposed parameterization.
>
> We will continue refining the manuscript to make this distinction clearer. We thank the reviewer again for the helpful comment.

---

> ### Author Response · Authors · 2025-12-01
> **Response for Weakness 3. Part(1/2)**
>
> Thank you for highlighting the need for a baseline that merges all retrieved documents into a single adaptation without orthogonal decomposition. We agree that such comparisons are important for isolating the contribution of the
> orthogonal merging mechanism. In the revised manuscript, we have added Section **4.3.3: Effect of the Merging Methods**, which includes ablation experiments evaluating several alternative merging strategies (Table\~5 and Table\~6).
>
> **Table 6: MergePRAG+ / HotpotQA — Performance comparison between different merging methods
> (Setting: $\|SP\|$=1)**
>
> | $\mathsf{Merge}_{seq}$       | EM    | F1    |
> |------------------------------|-------|-------|
> | $\blacktriangle$             | 36.20 | 43.47 |
> | $\bullet$                    | 48.20 | 55.04 |
> | $\blacklozenge$              | 46.40 | 54.07 |
> | $\blacktriangledown$         | 48.20 | 54.95 |
> | $\blacksquare$               | 48.80 | 55.53 |
>
> **Legend:**
> - $\blacktriangle$: Adding merging
> - $\bullet$: Arithmetic mean merging
> - $\blacklozenge$: TIES merging
> - $\blacktriangledown$: Connection merging
> - $\blacksquare$: Gram–Schmidt orthogonalization merging
>
> **Table 5: Performance comparison between different merging methods for $\mathsf{Merge}_{inner}$ and $\mathsf{Merge}\_{seq}$ varying \|SP\|**
>
> | $\mathsf{Merge}_{inner}$ | $\mathsf{Merge}_{seq}$ | \|SP\|=2 |     | \|SP\|=4 |     | \|SP\|=6 |     | \|SP\|=8 |     | \|SP\|=10 |     | \|SP\|=12 |     |
> |--------------------------|------------------------|---------|-----|---------|-----|---------|-----|---------|-----|----------|-----|----------|-----|
> |                          |                        | EM      | F1  | EM      | F1  | EM      | F1  | EM      | F1  | EM       | F1  | EM       | F1  |
> | $\bullet$                | $\bullet$               | 50.20 | 58.07 | 51.40 | 60.35 | 51.40 | 59.82 | 52.00 | 60.86 | 55.00 | 62.84 | 54.40 | 62.76 |
> | $\bullet$                | $\blacksquare$          | 50.60 | 58.06 | 52.00 | 60.64 | 52.00 | 60.21 | 53.00 | 61.32 | 55.40 | 63.40 | 54.60 | 62.64 |
> | $\blacksquare$           | $\bullet$               | 50.40 | 58.14 | 51.60 | 60.13 | 51.40 | 59.63 | 52.60 | 61.46 | 54.80 | 62.80 | 54.60 | 62.76|
> | $\blacksquare$           | $\blacksquare$          | **50.80** | **58.23** | **52.40** | **60.67** | **52.40** | **60.67** | **53.40** | **61.77** | **55.60** | **63.45** | **55.00** | **62.93** |
>
> **Legend:**
> - $\bullet$：Arithmetic mean merging
> - $\blacksquare$：Gram–Schmidt orthogonalization merging
>
> Specifically, we introduce three non-orthogonal baselines—additive merging, arithmetic mean merging, and concat merging—implemented as described in **Appendix~F: Merging Introduction**. These baselines fuse all retrieved passage-memory vectors into a single representation by summation, averaging, or concatenation, without applying any orthogonalization. Across all evaluated
> datasets, these non-orthogonal approaches consistently underperform our Gram–Schmidt orthogonalization–based merging.
>
> As shown in Table\~6, orthogonal merging achieves the best performance for sequential (inter-) merging, with arithmetic mean merging being somewhat competitive. However, Table~5 further demonstrates that orthogonal merging consistently yields the strongest results, often improving EM by more than 1\% over arithmetic pooling. These findings confirm that orthogonal merging
> is a key mechanism: it mitigates conflicts between retrieved passages and preserves complementary information, thereby enhancing the effectiveness of MergePRAG.

---

> ### Author Response · Authors · 2025-12-01
> **Response for Weakness 3. Part(2/2)**
>
> Regarding LoRA-based baselines, we performed preliminary experiments applying LoRA within a RAG setting. However, these combined runs were not competitive with MergePRAG+, which motivated our focus on hypernetwork-based injection. We did not include these preliminary results due to the need for rerunning them, but we plan to provide more extensive LoRA comparisons in an extended version.
>
> **It is also worth clarifying that, while LoRA is not used for passage-level parameterization in the main experiments, Table\~3 already compares full fine-tuning against our hypernetwork-based approach. In this comparison, ***MultiHopRAG with full fine-tuning*** substantially underperforms ***MergePRAG+***, demonstrating the advantage of the proposed hypernetwork-based parameterization.**
>
> **Table 3: Comparison of MergePRAG+ and MultihopRAG with fine-tuning (without hypernetwork) under LLaMA3.1-8B**
> | Training type                                        | HotpotQA EM | HotpotQA F1 | 2WikiMhQA EM | 2WikiMhQA F1 |
> |------------------------------------------------------|-------------|-------------|--------------|--------------|
> | MultihopRAG$_{\|SP\| = 1}$  (w/o finetuning)              | 37.80       | 47.56       | 23.30        | 35.59        |
> | MultihopRAG$_{\|SP\|=1}$ (finetuning: [sq → sa])       | 43.70       | 50.15       | 58.10        | 62.57        |
> | MultihopRAG$_{\|SP\|=1}$ (finetuning: [(P_golden, sq) → sa]) | 40.10 | 46.79       | 60.30        | 62.04        |
> | MergePRAG + $_{\|SP\|=1}$                              | 47.40       | 55.29       | 65.60        | 70.54        |
>
> Finally, we note that recent work[1] also finds that combining retrieval with parameterized knowledge is not straightforward:
>
> Their observations align with our motivation that simple fine-tuning or naïve parameter addition does not reliably improve RAG, reinforcing the need for principled mechanisms such as our orthogonal merging.
>
> We thank the reviewer again for this insightful comment and will continue to expand these comparisons in an extended version.
>
> ### **Reference**
>
> [1] Ovedia O, Brief M, Mishaeli M, et al. \textit{Fine-tuning or retrieval? Comparing knowledge injection in LLMs}. arXiv:2312.05934, 2023.

---

> ### Author Response · Authors · 2025-12-01
> **Response for Weakness 4.**
>
> Thank you for raising this concern. We clarify that the choice of “critical layers’’ in our method is not arbitrary. In **Appendix A: Layer Scanning Experiments for Critical-Layer Parameterization**, we provide systematic layer-scanning analyses for both LLaMA-3.1-8B and Qwen-2.5-7B across multiple datasets (**Figures 2–7**). The perplexity curves show clear and substantial variation across layers, and importantly, the optimal injection layer differs significantly between architectures. These results demonstrate that layer sensitivity is model-dependent, making empirical identification necessary rather than assuming a fixed layer a priori.
>
> While layer-by-layer scanning may appear inelegant, it is fully aligned with established practice in prior knowledge-editing work such as ROME[1] and MEMIT[2], which similarly rely on scanning (and in some cases token-level probing) to locate the most influential intervention point.
>
> Our approach is consistent with this “locate-then-edit’’ paradigm; in our setting, MergePRAG can be viewed as a corresponding ***locate-then-inject*** method.
>
> Our experiments also show that optimal layers are not shared across datasets or architectures, suggesting that a universal critical layer is unlikely to exist. A lightweight scanning procedure therefore provides a model-adaptive and data-driven solution. This procedure is efficient: our hypernetwork is small, and scanning requires only forward passes (e.g., perplexity evaluation), without
> additional training.
>
> Conceptually, identifying a critical layer is essential because different layers serve different representational roles. Injecting passage vectors too early may prevent effective use of lexical information, while injecting too late may prevent the injected knowledge from influencing reasoning. Scanning enables us to target the representational stage where external knowledge integrates most effectively.
>
> At the same time, we acknowledge that the optimal layer may vary across tasks. The current results are validated in QA-based RAG settings; other reasoning-intensive tasks (e.g., mathematical or symbolic reasoning) may require further analysis using more advanced causal or mechanistic interpretability methods, as discussed in recent work such as Biran E et al.[3]. We plan to explore these directions in future work.
>
> In summary, although scanning introduces a small procedural step, it is empirically justified, consistent with prior literature, computationally lightweight, and necessary to ensure effective parametric knowledge injection across diverse architectures and tasks. The proposed critical-layer selection is motivated by the widely adopted locate-then-edit framework, and MergePRAG
> naturally fits into this locate-then-inject perspective. We will continue to refine and extend our analysis in future iterations. Thank you again for this valuable comment.
>
> ### **Reference**
> [1]Meng K, Bau D, Andonian A, et al. Locating and editing factual associations in gpt[J]. Advances in neural information processing systems, 2022, 35: 17359-17372.
> [2] Meng K, Sharma A S, Andonian A J, et al. Mass-Editing Memory in a Transformer[C]//ICLR. 2023.
> [3] Biran E, Gottesman D, Yang S, et al. Hopping too late: Exploring the limitations of large language models on multi-hop queries[J]. arXiv preprint arXiv:2406.12775, 2024.

---

> ### Author Response · Authors · 2025-12-01
> **Response for Weakness 5.**
>
> Thank you for raising this concern. While it is not feasible to evaluate MergePRAG on all possible model architectures, we selected two structurally distinct and widely used backbones—LLaMA3.1-8B and Qwen2.5-7B—to assess the method’s generality. The consistent improvements observed across these models suggest that the core components of MergePRAG -- hypernetwork-based
> parameterization, orthogonal merging, and critical-layer scanning -- are architecture-agnostic and can be applied broadly to Transformer-based models.
>
> In particular, because the hypernetwork is small and scanning relies only on forward-pass evaluations (e.g., perplexity), this
> procedure remains efficient and practical even for larger models.
>
> Of course, we do not claim that MergePRAG will work uniformly for all models. Rather, we would like to present that the positive results on two moderately sized models indicate promising generalization potential.
>
> As we are very curious about how MergePRAG performs on other types of models, we are interested in extending our experiments to additional backbones, including larger models such as LLaMA-30B, as computational resources become available.
> We appreciate the reviewer’s comment and will continue to expand our evaluation to further validate the generality of MergePRAG.

---

### Official Review · Reviewer_xajy · 2025-10-31

**Soundness:** 3
**Presentation:** 2
**Contribution:** 2
**Rating:** 4
**Confidence:** 4

**Summary:**

The paper proposes extending Parametric RAG (PRAG) to handle multi-hop question answering by decomposing questions into atomic sub-questions and recursively merging a hypernetwork with retrieved passages.

**Strengths:**

- The problem addressed in this paper is well-motivated: adapting Parametric RAG (PRAG) for multi-hop question answering.
- The paper decomposes a multi-hop question into atomic sub-questions and recursively merges the hypernetwork with each retrieved passage.
- The proposed method extends prior work, enabling PRAG to effectively handle multi-hop reasoning tasks.

**Weaknesses:**

- Baseline selection and setup:
  - The baseline selection is weak. Some competitive baselines are missing, such as Adaptive-RAG [1] for (i) RAG, and Search-o1 [2] and Search-R1 [3] for (ii) iterative retrieval.
  - The experimental settings for PRAG should be comparable to those of the proposed method. MergePRAG may gain advantages from sub-question decomposition and sub-answer generation. Even though PRAG was not originally designed for multi-hop retrieval, the sub-question decomposition should still be applied to the plain PRAG for fair comparison.

- Missing ablation study:
  - The paper does not include ablation studies on individual components (e.g., orthogonal merge, critical layer parameterization). Such analyses would help readers understand the necessity and novelty of each component.

- Additional resource considerations:
  - The proposed method requires sub-question decomposition and sub-answer generation, each followed by a hypernetwork merging step.    However, the inference cost (e.g., number of inferences for sub-answer generation or latency) is not compared. Such an analysis is needed to confirm that the performance gains do not come at the expense of computational efficiency.

[1] Jeong et al., "Adaptive-RAG: Learning to Adapt Retrieval-Augmented Large Language Models through Question Complexity", NAACL 2024.               \
[2] Li et al., "Search-o1: Agentic Search-Enhanced Large Reasoning Models", ArXiv 2025.
[3] Jin et al., "Search-R1: Training LLMs to Reason and Leverage Search Engines with Reinforcement Learning", ArXiv 2025.

**Questions:**

This paper iteratively merges the hypernetwork using retrievals from each step’s sub-question and generates sub-answers, denoted in Eq. (4) as:

$$
s_{a_t} = M_{\theta_0 \oplus F(SP_{1:t})}(sq_t)
$$

However, why not:
1. Generate a single hypernetwork using the passages from all steps at once,
   $$
   H(SP_{1:T})
   $$
   instead of sequentially?
2. Or generate sub-answers using the fully merged hypernetwork,
   $$
   s_{a_t} = M_{\theta_0 + F(SP_{1:T})}(sq_t)
   $$

---

> ### Author Response · Authors · 2025-12-01
> **Response for Weakness 1.**
>
> Thank you for raising this concern.
>
> 1. To strengthen the baseline selection, we have added **Table~15** in the revised manuscript, which reports comparisons against Adaptive-RAG, Search-o1, and Search-R1. Although Adaptive-RAG is implemented on a different pretrained backbone (FLAN-T5-XL), we include it for completeness. MergePRAG+ shows clear improvements over Search-o1 and Search-R1 on all three benchmarks. Compared with Adaptive-RAG, MergePRAG+ achieves higher EM on HotpotQA and outperforms it on 2WikiMhQA, while performing slightly lower on MuSiQue. We also note that MergePRAG+ can serve as a complementary backbone and may be combined with Adaptive-RAG or other advanced RAG approaches—an interesting direction for future exploration.
>
> **Table 15: Performance comparison of MergePRAG+ with other advanced RAG methods on three QA benchmarks -- Adaptive-RAG, Search-R1 and Search-o1.**
>
> | Model        | Method                  | Retriever | HotpotQA EM | HotpotQA F1 | 2WikiMhQA EM | 2WikiMhQA F1 | MuSiQue EM | MuSiQue F1 |
> |:------------:|:----------------------:|:---------:|:-----------:|:-----------:|:------------:|:------------:|:----------:|:-----------:|
> | FLAN-T5-XL   | Adaptive-RAG            | BM25      | 42.00       | 53.82       | 40.60        | 49.75        | 23.60      | 31.80       |
> | LLaMA3.1-8B  | Search-o1               | BM25      | 14.80       | 24.08       | 22.20        | 27.10        | 5.40       | 11.98       |
> | Qwen2.5-7B   | Search-o1               | BM25      | 11.60       | 16.95       | 22.00        | 25.02        | 2.10       | 7.48        |
> | Qwen2.5-3B   | Search-R1               | E5        | 32.40       | -           | 31.90        | -            | 10.30      | -           |
> | Qwen2.5-7B   | Search-R1               | E5        | 37.00       | -           | 41.40        | -            | 14.60      | -           |
> | Qwen2.5-7B   | MergePRAG$_{\|SP\|=1}$    | E5        | 43.40       | 50.64       | 65.80        | 69.72        | 9.70       | 19.61       |
> | Qwen2.5-7B   | MergePRAG$_{\|SP\|=1}$    | BM25      | 42.00       | 49.09       | 59.70        | 63.05        | 13.00      | 23.35       |
> | LLaMA3.1-8B  | MergePRAG$_{\|SP\|=1}$    | E5        | 48.80       | 55.53       | 66.30        | 71.05        | 14.40      | 25.04       |
> | LLaMA3.1-8B  | MergePRAG$_{\|SP\|=1}$    | BM25      | 46.80       | 53.40       | 61.60        | 67.31        | 17.80      | 29.39       |
>
>
> 2. Regarding the experimental settings for PRAG, we agree that a fair comparison requires controlling for the effects of sub-question decomposition. To this end, we include the MultiHopRAG baseline, which applies the same sub-question decomposition module together with RAG-style fine-tuning. This baseline can be viewed as a PRAG variant equipped with decomposition. As shown in **Table\~3**, even when all methods use the identical sub-question generator, MergePRAG achieves the strongest performance. This demonstrates that the improvements do not arise solely from decomposition. Rather, the key advantage of our method comes from the passage-parameterization mechanism, which provides explicit parametric knowledge integration beyond what PRAG or its decomposition-augmented variant can offer.
>
> **Table 3: Comparison of MergePRAG+ and MultihopRAG with fine-tuning (without hypernetwork) under LLaMA3.1-8B**
> | Training type                                        | HotpotQA EM | HotpotQA F1 | 2WikiMhQA EM | 2WikiMhQA F1 |
> |------------------------------------------------------|-------------|-------------|--------------|--------------|
> | MultihopRAG$_{\|SP\| = 1}$  (w/o finetuning)              | 37.80       | 47.56       | 23.30        | 35.59        |
> | MultihopRAG$_{\|SP\|=1}$ (finetuning: [sq → sa])       | 43.70       | 50.15       | 58.10        | 62.57        |
> | MultihopRAG$_{\|SP\|=1}$ (finetuning: [(P_golden, sq) → sa]) | 40.10 | 46.79       | 60.30        | 62.04        |
> | MergePRAG + $_{\|SP\|=1}$                              | 47.40       | 55.29       | 65.60        | 70.54        |
>
> We thank the reviewer again for these constructive suggestions.

---

> ### Author Response · Authors · 2025-12-01
> **Response for Weakness 2.**
>
> Thank you for the suggestion. We have added a new **section, 4.3.3: Effect of the Merging Strategy**, which provides ablation experiments comparing orthogonal merging with several alternative merging strategies (Table\~5 and Table\~6). The results clearly demonstrate that orthogonal merging is essential for effectively combining paragraph vectors.
>
> **Table 6: MergePRAG+ / HotpotQA — Performance comparison between different merging methods
> (Setting: $\|SP\|$=1)**
>
> | $\mathsf{Merge}_{seq}$       | EM    | F1    |
> |------------------------------|-------|-------|
> | $\blacktriangle$             | 36.20 | 43.47 |
> | $\bullet$                    | 48.20 | 55.04 |
> | $\blacklozenge$              | 46.40 | 54.07 |
> | $\blacktriangledown$         | 48.20 | 54.95 |
> | $\blacksquare$               | 48.80 | 55.53 |
>
> **Legend:**
> - $\blacktriangle$: Adding merging
> - $\bullet$: Arithmetic mean merging
> - $\blacklozenge$: TIES merging
> - $\blacktriangledown$: Connection merging
> - $\blacksquare$: Gram–Schmidt orthogonalization merging
>
> **Table 5: Performance comparison between different merging methods for $\mathsf{Merge}_{inner}$ and $\mathsf{Merge}\_{seq}$ varying \|SP\|**
>
> | $\mathsf{Merge}_{inner}$ | $\mathsf{Merge}_{seq}$ | \|SP\|=2 |     | \|SP\|=4 |     | \|SP\|=6 |     | \|SP\|=8 |     | \|SP\|=10 |     | \|SP\|=12 |     |
> |--------------------------|------------------------|---------|-----|---------|-----|---------|-----|---------|-----|----------|-----|----------|-----|
> |                          |                        | EM      | F1  | EM      | F1  | EM      | F1  | EM      | F1  | EM       | F1  | EM       | F1  |
> | $\bullet$                | $\bullet$               | 50.20 | 58.07 | 51.40 | 60.35 | 51.40 | 59.82 | 52.00 | 60.86 | 55.00 | 62.84 | 54.40 | 62.76 |
> | $\bullet$                | $\blacksquare$          | 50.60 | 58.06 | 52.00 | 60.64 | 52.00 | 60.21 | 53.00 | 61.32 | 55.40 | 63.40 | 54.60 | 62.64 |
> | $\blacksquare$           | $\bullet$               | 50.40 | 58.14 | 51.60 | 60.13 | 51.40 | 59.63 | 52.60 | 61.46 | 54.80 | 62.80 | 54.60 | 62.76|
> | $\blacksquare$           | $\blacksquare$          | **50.80** | **58.23** | **52.40** | **60.67** | **52.40** | **60.67** | **53.40** | **61.77** | **55.60** | **63.45** | **55.00** | **62.93** |
>
> **Legend:**
> - $\bullet$：Arithmetic mean merging
> - $\blacksquare$：Gram–Schmidt orthogonalization merging
>
> For critical layer selection, we provide detailed analysis in **Appendix A: Layer Scanning Experiments for Critical Layer Parameterization**. These experiments show that inserting paragraph vectors at layers with low perplexity leads to the most effective responses, confirming the necessity of identifying the optimal critical layer for knowledge injection.
>
> Together, these analyses isolate and validate the contribution of each component, supporting both the novelty and effectiveness of our approach.

---

> ### Author Response · Authors · 2025-12-01
> **Response for Weakness 3.**
>
> Thank you for raising this important point. In response to your comment, we have added a new section, **Appendix D.3: Efficiency Analysis**, where we provide a detailed breakdown of the inference-time cost, including paragraph-vector generation, sub-question decomposition, sub-answer generation, and the overall end-to-end latency. (Table\~11)
>
> **Table 11: Efficiency analysis of MergePRAG using the LLaMA3.1-8B model on the HotpotQA dataset**
> | Method | Generate Passage Memory | Response $sa$ | Average Time |
> |:------:|:---------------------:|:------------:|:------------:|
> | $\text{RAG}_{\|SP\|=1}$ | - | - | 0.712s |
> | $\text{RAG-CoT}_{\|SP\|=1}$ | - | - | 6.389s |
> | $\text{MergePRAG+}_{\|SP\|=1}$ | 0.001s | 0.259s | 2.517s |
>
> Our analysis shows that the overhead introduced by MergePRAG+ is not prohibitive for several reasons:
>
> (a) **Lightweight hypernetwork.**
> The hypernetwork is extremely small, so generating paragraph vectors is fast and contributes negligible latency relative to the base model forward pass.
>
> (b) **Low-cost sub-answer generation.**
> Although sub-answer generation introduces additional inference steps, each step
> operates on a short, simplified sub-question, keeping per-step cost low.
>
> (c) **Acceptable total inference time.**
> Question decomposition increases the number of reasoning steps compared to standard RAG, but the overall latency remains well within practical limits, especially given the substantial performance improvements obtained by MergePRAG+ for multi-hop QA. It is also worth noting that such decomposition is commonly employed by advanced RAG methods that rely on reasoning or reflection chains. For example, recent work [1] also adopts query decomposition to enable effective multi-hop question answering.
>
> Overall, the efficiency analysis confirms that the gains of MergePRAG+ do not come at the expense of excessive computational cost. The additional steps introduced by MergePRAG+ are justified by the improvements over both standard RAG and our non-parametric baseline MergePRAG. We will continue refining the efficiency discussion for clarity in future revisions.
>
> Thank you again for this valuable comment.
>
> ### **Reference**
> [1] Zhao S, Yu T, Xu A, et al. Parallelsearch: Train your llms to decompose query and search sub-queries in parallel with reinforcement learning[J]. arXiv preprint arXiv:2508.09303, 2025.

---

> ### Author Response · Authors · 2025-12-01
> **Response for Question 1.**
>
> Thank you for raising these questions. We agree that the suggested alternatives:
>
> (i) generating a single hypernetwork using all retrieved passages at once, or
> (ii) generating all sub-answers using the fully merged hypernetwork
>
> are interesting baselines, and we plan to include such comparisons in future extensions. Below we clarify why our current design
> adopts the sequential merging approach.
>
> **1. Unified hypernetwork over all passages.**
> A unified hypernetwork that ingests all retrieved passages at once would require concatenating all passages across hops into a single long sequence.
>
> This would substantially increase the semantic and computational burden on the hypernetwork, which would then need to simultaneously perform ***parameterization*** and ***merging*** internally. Our design philosophy explicitly separates these two responsibilities: the hypernetwork is kept lightweight and encodes only a single passage at a time, while the merging
> module is responsible for integrating multiple passage experts. This modular separation is key to ensuring scalability and interpretability. Designing a hypernetwork that jointly performs parameterization and merging is indeed an interesting future direction.
>
> **2. Using the fully merged hypernetwork for all sub-answers.**
> Our method assumes a multi-hop retrieval setting in which passages arrive incrementally, and each sub-question $sq_t$ is typically most strongly supported by the passages retrieved at step $t$. Using the fully merged representation for all sub-answers may dilute this step-specific relevance, since the merged memory would also contain information from passages retrieved for later sub-questions. Sequential merging preserves the ability to respond to each sub-question using the most relevant passage-level information available at that moment, while still cumulatively encoding earlier evidence in a stable manner through orthogonal merging.
>
> At this stage, we believe that restricting the passage input for each sub-answer to the passages retrieved at that hop is a principled and performance-aligned choice. Nonetheless, we acknowledge that using the full set of retrieved passages for all sub-answers could also have advantages, especially given the logical connections among sub-questions. Inspired by your suggestion, we plan to explore unified-passage variants and include these results in an extended version.
>
> We appreciate the reviewer’s insightful question, which has opened an interesting avenue for further experimentation.

---

### Meta-Review · Area_Chair_SJqV · 2026-01-07

**Summary:**

The paper proposes MergePRAG to extend the Parametric RAG in the multi-hop retrieval setting. It first decomposes questions into atomic sub-questions and recursively integrates retrieved passages into LLM parameters through orthogonal merging. All reviewers agree that the proposed framework aims to deal with an important problem, and the framework is novel. There are some concerns regarding with insufficient ablation studies, missing several important baselines and models.

**Reviewer Concerns:**

The rebuttal solves the reviewers' concerns by adding a detailed ablation study to show each component's importance. It also includes several baselines to make the experiments more comprehensive.

**Reviewer Scores:**

I believe the rebuttal provides a lot of new experimental results on new baselines, and ablation studies help to solve the reviewer's concern. I expect Reviewer xajy will increase the score to 6, and Reviewer EL8F will increase the score to 4.

---

### Decision · Program_Chairs · 2026-01-26

Accept (Poster)